# A Comparison and Evaluation of Stereo Matching on Active Stereo Images

**DOI:** 10.3390/s22093332

**Published:** 2022-04-26

**Authors:** Mingyu Jang, Hyunse Yoon, Seongmin Lee, Jiwoo Kang, Sanghoon Lee

**Affiliations:** 1Department of Electrical and Electronic Engineering, Yonsei University, Seoul 03722, Korea; jmg1002@yonsei.ac.kr (M.J.); hsyoon97@yonsei.ac.kr (H.Y.); lseong721@yonsei.ac.kr (S.L.); 2Department of IT Engineering, Sookmyung Women’s University, Seoul 04310, Korea; 3Department of Radiology, College of Medicine, Yonsei University, Seoul 03722, Korea

**Keywords:** active stereo matching, off-the-shelf active stereo sensor, matching cost, performance evaluation, infrared image, disparity accuracy

## Abstract

The relationship between the disparity and depth information of corresponding pixels is inversely proportional. Thus, in order to accurately estimate depth from stereo vision, it is important to obtain accurate disparity maps, which encode the difference between horizontal coordinates of corresponding image points. Stereo vision can be classified as either passive or active. Active stereo vision generates pattern texture, which passive stereo vision does not have, on the image to fill the textureless regions. In passive stereo vision, many surveys have discovered that disparity accuracy is heavily reliant on attributes, such as radiometric variation and color variation, and have found the best-performing conditions. However, in active stereo matching, the accuracy of the disparity map is influenced not only by those affecting the passive stereo technique, but also by the attributes of the generated pattern textures. Therefore, in this paper, we analyze and evaluate the relationship between the performance of the active stereo technique and the attributes of pattern texture. When evaluating, experiments are conducted under various settings, such as changing the *pattern intensity, pattern contrast, number of pattern dots*, and *global gain*, that may affect the overall performance of the active stereo matching technique. Through this evaluation, our discovery can act as a noteworthy reference for constructing an active stereo system.

## 1. Introduction

Estimating an accurate depth from stereo pair images is a critical problem for achieving high-performance 3D applications in the field of computer vision [1,2,3,4,5]. In order to obtain depth from stereo vision, a disparity map should be estimated. A disparity map is a two-dimensional (2D) map that marks the pixel position difference between matching points of two images. Because the accuracy of depth is dependent on the accuracy of disparity, estimating an accurate disparity map is crucial in stereo vision [6].

Stereo vision can be categorized into two types: passive and active. The difference between the passive and active methods is the existence of a pattern projector. The passive stereo technique matches pixel correspondences using image features between two RGB images without having the pattern texture projected from the projector onto the scene [7,8,9,10,11]. These image features of a scene are influenced by the complexity of the scene structure, as well as by the passive texture, which is affected by light conditions. Thus, unique features may not be captured in the textureless regions, so the probability of making false matches increases due to the incapability of the technique to differentiate between the features of different sources. As shown in Figure 1, the disparity map generated from passive stereo vision shows many errors on the ground, curtain, and mannequin, which are textureless regions. On the other hand, the active stereo techniques are supplemented by a projector that emits a patterned texture onto the surface of an object [12]. Generally, active stereo techniques use an infrared (IR) pattern projector and an IR sensor to be reliable against visible light interference such as background light sources.

Thus, active stereo techniques accurately match the corresponding pixels between stereo images, even for textureless areas of objects. As a result, Figure 1 illustrates that the errors on the textureless regions are removed from the disparity map generated from the active stereo vision. Generally, active stereo techniques use an infrared (IR) pattern projector and an IR sensor to be reliable against visible light interference such as background light sources.

Many surveys [13,14,15,16,17] have been conducted to evaluate the performance of many matching techniques on passive stereo images. Previous surveys’ experiments and benchmarks have analyzed the interaction between attributes affecting the passive texture and the performance of the passive stereo techniques. However, unlike the passive stereo technique, the performance of active stereo techniques is affected by both passive texture and pattern texture. Because previous surveys only evaluate a fraction of attributes affecting the performance of the active stereo technique, the relationship between the performance of the active stereo technique and the pattern attributes is still unknown.

Deep learning-based approaches may outperform traditional methods in many cases. However, the performance of deep learning-based methods is highly dependent on the dataset used to train the model. As a result, if the data given does not resemble any data in the training dataset, the performance of the deep learning-based method would most likely be low. Thus, the performance of the deep learning-based method is heavily dependent on the environment. In addition, since the deep learning-based method is dependent on the training data, it is difficult to analyze whether the performance evaluation result according to the pattern attributes is over-fitting the learning data or is dependent on the pattern attributes. In contrast, the traditional methods can be applied and analyzed generally.

Thus, in this paper, we evaluate the relationship between the attributes of pattern texture, namely, the *pattern intensity, pattern contrast, number of pattern dots*, and active stereo matching performance. For a quantitative evaluation, we acquired a synthetic active stereo dataset with a ground-truth, which allows each attribute of pattern texture to be independently controlled. The synthetic active stereo dataset is generated by adding the synthetic pattern texture, which mimics the pattern texture projected by the real projector, onto the public passive stereo dataset. Our evalutaion used not only the synthetic dataset, but also the real dataset. The real dataset used is captured by the an off-the-shelf active stereo camera. The attributes of pattern texture depend on the settings of the active stereo camera. Fortunately, most off-the-shelf active stereo matching cameras allow users to control many attributes of pattern texture, such as the intensity of pattern. Through testing on the real dataset, we confirm that our evaluation is valid in the real-world environment, and ensure that it will provide guidance to construct an active stereo sensor system. In addition, regardless of the dataset, our experimental results on the pattern attributes that affect the IR environment can provide guidance for constructing an active stereo sensor system for many applications of active stereo matching techniques.

## 2. Related Works

### 2.1. Active Stereo Matching

Active stereo matching computes the disparities between coordinates of corresponding pixels in IR stereo images containing projected IR patterns. Images are first rectified by changing them such that their epipolar lines are horizontal and have the same y-coordinate. This change causes corresponding pixels to have the same y-coordinates, which reduces the disparity computation from a 2D space to a one-dimensional (1D) space, which reduces computational costs [6].

After the images are rectified, the matching cost is calculated, the costs are aggregated, the disparities are actually calculated, and the matches are refined. Matching costs are the errors that occur when the wrong pixels are identified as being corresponding ones. Matching costs are computed using either parametric or non-parametric methods. The matching costs computed for each disparity are aggregated in a local window and are then used to find the correct disparity. After the disparity is selected, the computed disparity is refined.

The evaluated combinations of matching costs and stereo algorithms are similar to those used in other passive stereo image studies [13,14]. The parametric matching costs used in this study were classified as either pixel-wise or window-based costs. The pixel-wise parametric methods used in this study were the absolute differences (AD) and the Birchfield and Tomasi (BT) methods [18]. The window-based methods were the sum of absolute differences (SAD), the zero-mean sum of absolute differences (ZSAD), normalized cross-correlation (NCC), and zero-mean normalized cross-correlation (ZNCC) methods. Instead of computing matching costs based on raw data, we applied the mean, the Laplacian of Gaussian [9], bilateral filters [19], and a rank filter to remove image offsets. Non-parametric methods are based on the ordering of intensities, so they are robust against outliers near object boundaries [20]. Ordering also improves robustness against global gains because the ordering of the intensities remains consistent. In this paper, we used Census [21], which is the most commonly used non-parametric matching cost.

For benchmarking, the commonly used stereo matching algorithms are window-based [9], semi-global [10], and global methods using graph-cuts [11]. The window-based method aggregates the matching cost by summing or averaging costs over a window, and selects a disparity of the lowest cost. Global methods, such as graph-cuts, use an energy function to minimize energy in a 2D space globally. The semi-global method works similarly to the global method by utilizing the energy function to minimize the energy. However, instead of collectively minimizing the energies in the entire 2D space, the semi-global method minimizes the energy along the 1D path, which runs toward the pixel of interest.

### 2.2. Active Stereo Sensor

Because of accelerating research in computer vision, many off-the-shelf RGB-D cameras are commercially available at affordable prices. The most commonly used commercial RGB-D cameras are the Microsoft Kinect Azure (Microsoft, Redmond, WA, USA) [22], the Orbbec Astra series (Orbbec, Troy, MI, USA) [23], and the Intel RealSense series D400 [24] and L515 (Intel, Santa, Clara, CA, USA) [25]. These RGB-D cameras use different technologies to measure depth. The Microsoft Kinect Azure uses the time-of-flight (ToF) [26] technique to obtain the depth map. The Microsoft Kinect Azure consists of an emitter and a receiver, which are used to measure the round-trip time when a signal from an emitter returns to the receiver. Based on the measured round-trip time, the device estimates the depth. Unlike the Microsoft Kinect Azure, the Orbbec Astra series uses a structured light technique to estimate depth. In the structured light technique, patterns whose original shapes are known in advance are projected to a target object. The depth is estimated using geometric relationships between the original and deformed pattern shapes. The Intel RealSense L515 uses a light detection and ranging (LiDAR) technique, one of the ToF techniques. The LiDAR system emits an IR laser that hits a target before being reflected back to a sensor located close to the light source. By measuring the time taken for the light to travel, and knowing the constant speed of light, the target’s distance can be calculated with a high degree of accuracy. The Intel RealSense D400 series cameras are the only cameras that utilize the active stereo technique to estimate depth among the mentioned cameras. The Intel Realsense D400 series has a projector, which emits an unstructured pattern to add texture to the object’s surface. At the same time, the depth is calculated by matching texture correspondences between images captured by two infrared (IR) cameras

Table 1 summarizes the specification of these Intel RealSense D400 series cameras. Even though the Intel RealSense D415, D435, and D455 series cameras all use the active stereo technique, the pattern used by each model is different. The D415 uses two AMS Heptagon projectors, while the D435 and D455 use an AMS Princeton Optronics projector with a wider emission angle but fewer spots. Because the field of view (FOV) of the RealSense D415 model is narrower than the RealSense D435 and D455 models, the patterns are more densely packed. Nevertheless, the pattern density for RealSense D435 and D455 can increase by increasing the number of the same projector. In addition, the AMS Heptagon projector projects a pattern with a specific structure. On the other hand, the AMS Princeton Optronics projector randomly projects patterns. Thus, in terms of simulating the pattern projection on the synthetic dataset, it is easier to simulate AMS Princeton Optronics projector than AMS Heptagon projector. Furthermore, RealSense D455 has the longest baseline, which allows this model to estimate depth with higher accuracy.

Therefore, in this paper, we used the RealSense D455 model for our experiments. In addition, we synthetically mimicked the pattern projected by the RealSense D455 and applied it onto the public passive stereo dataset Middlebury [27] so that we could evaluate the matching cost and stereo algorithms on the synthetically generated active stereo dataset.

## 3. Method

### 3.1. Preprocessing Filter

#### 3.1.1. Mean Filter

The mean filter removes the local intensity offset by subtracting each pixel from the mean intensities within a window centered at the pixel of interest, which is defined as:(1)$$\begin{array}{c} I_{\mathrm{mean}}\left(p\right)=I\left(p\right)-\frac{1}{\left|N_{p}\right|}\sum_{q\in N_{p}}I\left(q\right)+128. \end{array}$$

After subtracting the pixel, the constant offset of 128 is added to the difference in order to avoid negative numbers when storing the result back into an 8-bit image. Unlike ZSAD, each pixel has its window for computing the mean intensity of its neighboring pixels.

#### 3.1.2. Laplacian of Gaussian Filter

The Laplacian of Gaussian (LOG) [9] is a bandpass filter used to remove noise and offsets in intensities by performing smoothing. The LoG filter is defined as:(2)$$\begin{array}{c} \begin{array}{cc} I_{\mathrm{LoG}} & =I⊗K_{\mathrm{LoG}},\\ K_{\mathrm{LoG}}\left(x,y\right) & =-\frac{1}{\pi \sigma^{4}}\left(1-\frac{x^{2}+y^{2}}{2\sigma^{2}}\right)e^{\frac{x^{2}+y^{2}}{2\sigma^{2}}}, \end{array} \end{array}$$
where *x* and *y* are pixel coordinates and σ is the smoothing parameter.

#### 3.1.3. Bilateral Filtering

The bilateral filter [19] smooths images by removing local offsets without eliminating the details of high-contrast textures. Removing local offsets reduces the chance of depth discontinuities occurring. The bilateral filter works by assigning weights to pixels neighboring the target pixel based on their proximity and color similarity to the target pixel. Then, the intensity values of the target pixel and its neighbors are summed. The filtered image is then subtrated from the original image to remove the original image’s background. The bilateral filter is defined as:(3)$$\begin{array}{c} \begin{array}{cc} I_{\mathrm{BilSub}}\left(p\right) & =I\left(p\right)-\frac{\Sigma_{q\in N_{p}}I\left(q\right)e^{s}e^{r}}{\Sigma_{q\in N_{p}}e^{s}e^{r}},\\ s & =-\frac{{\left(q-p\right)}^{2}}{2\sigma_{s}^{2}},\\ r & =-\frac{{\left(I\left(q\right)-I\left(p\right)\right)}^{2}}{2\sigma_{r}^{2}}, \end{array} \end{array}$$
where $\sigma_{s}$ is a spatial distance and $\sigma_{r}$ is the radiometric distance. While the term *s* smooths the pixel value, the term *r* prevents over-smoothing over high-contrast textures by referring to the absolute difference between a neighboring pixel q and the center pixel p.

#### 3.1.4. Rank Filter

The rank filter is used to increase robustness against outliers within a window. It replaces pixel intensity with its intensity rank within the window. It is defined as:(4)$$\begin{array}{c} I_{\mathrm{Rank}}\left(p\right)=\sum_{q\in N_{p}}T\left[I\left(q\right)<I\left(p\right)\right] \end{array}$$
where T[·] is a conditional function that returns 1 if its argument is true, and 0 otherwise. Filtered pixel values depend on the intensity rank, so the rank filter is robust against radiometric distortions because they do not affect the intensity rank. However, replacing the pixel intensity with intensity rank causes blurring around object borders, which, in most cases, have depth discontinuities.

### 3.2. Matching Cost

The introduced matching costs can be classified into two categories: parametric and non-parametric. Parametric matching costs use the magnitude of pixel values, while non-parametric costs use only the local ordering of intensities and can, therefore, handle all monotonic mappings.

#### 3.2.1. Absolute Difference

One of the parametric costs evaluated in this paper is *absolute difference (AD)*, which is defined as:(5)$$\begin{array}{c} C_{\mathrm{AD}}\left(p,d\right)=\left|I_{L}\left(p\right)-I_{R}\left(p-d\right)\right| \end{array}$$
where p is a pixel in the left image, and d is a disparity between the left image $I_{L}$ and the right image $I_{R}$. AD simply uses an intensity difference between corresponding pixels in the left and right images because it assumes that corresponding pixels will have the same brightness.

While AD globally computes matching cost, the sum of absolute difference (SAD) computes local matching cost within a window, and is defined as:(6)$$\begin{array}{c} C_{\mathrm{SAD}}\left(p,d\right)=\sum_{q\in N_{p}}\left|I_{L}\left(q\right)-I_{R}\left(q-d\right)\right| \end{array}$$
where q is a pixel inside a neighborhood $N_{p}$ of pixels p.

#### 3.2.2. Zero-Mean Sum of Absolute Differences

The zero-mean sum of absolute difference (ZSAD) works very similarly to SAD, except the neighboring pixel q is subtracted by the mean intensity inside a window. ZSAD is defined as:(7)$$\begin{array}{c} \begin{array}{cc} C_{\mathrm{ZSAD}}\left(p,d\right) & =\sum_{q\in N_{p}}\left|\left(I_{L}\left(q\right)-{\bar{I}}_{L}\left(p\right)\right)-\left(I_{R}\left(q-d\right)-{\bar{I}}_{R}\left(p-d\right)\right)\right|,\\ {\bar{I}}_{L}\left(p\right) & =\frac{1}{\left|N_{p}\right|}\sum_{q\in N_{p}}I_{L}\left(q\right), \end{array} \end{array}$$
where ${\bar{I}}_{L}\left(p\right)$ and ${\bar{I}}_{R}\left(p\right)$ are the means of pixels inside a window $N_{p}$ for $I_{L}$ and $I_{R}$, respectively. When computing the cost, all pixels inside a window are subtracted from the same mean intensity.

#### 3.2.3. Birchfield–Tomasi

The Birchfield and Tomasi (BT) [18] cost measures the sampling-insensitive absolute difference between stereo images, and is defined as:(8)$$\begin{array}{c} \begin{array}{cc} C_{\mathrm{BT}}\left(p,d\right) & =\min\left(A,B\right),\\ A & =\max\left(0,I_{L}\left(p\right)-I_{R}^{max}\left(p-d\right),I_{R}^{min}\left(p-d\right)-I_{L}\left(p\right)\right),\\ B & =\max\left(0,I_{L}\left(p\right)-I_{R}^{max}\left(p-d\right),I_{R}^{min}\left(p-d\right)-I_{L}\left(p\right)\right),\\ I^{min}\left(p\right) & =\min\left(I^{-}\left(p\right),I\left(p\right),I^{+}\left(p\right)\right),\\ I^{max}\left(p\right) & =\max\left(I^{-}\left(p\right),I\left(p\right),I^{+}\left(p\right)\right),\\ I^{-}\left(p\right) & =\left(I\left(p-{\left[1\;0\right]}^{T}\right)+I\left(p\right)\right)/2,\\ I^{+}\left(p\right) & =\left(I\left(p+{\left[1\;0\right]}^{T}\right)+I\left(p\right)\right)/2. \end{array} \end{array}$$

While AD directly computes the difference between left and right images, BT computes the absolute distance between the left image and the extrema of a linearly interpolated pixel of interest in the right image.

#### 3.2.4. Normalized Cross-Correlation

Normalized cross-correlation (NCC) is a matching cost that can be applied only to window-based stereo algorithm. It is defined as:(9)$$\begin{array}{c} C_{\mathrm{NCC}}\left(p,d\right)=1-\frac{\sum_{q\in N_{p}}I_{L}\left(q\right)I_{R}\left(q-d\right)}{\sqrt{\sum_{q\in N_{p}}I_{L}{\left(q\right)}^{2}\sum_{q\in N_{p}}I_{R}{\left(q-d\right)}^{2}}}. \end{array}$$

NCC normalizes pixels inside a window $N_{p}$ centered at the pixel p. For each normalized pixel q in $N_{p}$, the cross-correlation is computed to measure the degree to which pixels in the left and right images correspond. If the pixel *q* in the left image corresponds to the pixel q−d at disparity level d in the right image, the value of NCC equals 0. Because NCC normalizes pixels, NCC is robust to gain changes and Gaussian noise. However, it also blurs depth discontinuities more than other matching costs, due to outliers.

#### 3.2.5. Zero-Mean Normalized Cross-Correlation

Similar to NCC, ZNCC computes costs using the cross-correlation between normalized pixels in a given window. The only difference between them is that ZNCC subtracts each pixel’s intensity from the mean intensity before computing the cost. Hence, ZNCC is defined as:(10)$$\begin{array}{c} C_{\mathrm{ZNCC}}\left(p,d\right)=\frac{\sum_{q\in N_{p}}\left(I_{L}\left(q\right){\bar{I}}_{L}\left(p\right)\right)\left(I_{R}\left(q-d\right)-{\bar{I}}_{R}\left(p-d\right)\right)}{\sqrt{\sum_{q\in N_{p}}{\left(I_{L}\left(q\right)-{\bar{I}}_{L}\left(p\right)\right)}^{2}\sum_{q\in N_{p}}{\left(I_{R}\left(q-d\right)-{\bar{I}}_{R}\left(p-d\right)\right)}^{2}}}. \end{array}$$

ZNCC is the only parametric cost that can compensate for differences in gains and offset within the correlation window.

#### 3.2.6. Census

Census [21] is a non-parametric matching cost, which is based on the local order of intensities. Census defines a bit string where each bit corresponds to a certain pixel in the local neighborhood of the pixel of interest. A bit is set to one of the corresponding pixels that has a higher intensity than the pixel of interest, and vice versa. Thus, the census cost stores both the intensity rank and the spatial structure of the local neighborhood. Transformed images are matched by computing the Hamming distance between corresponding bit strings.

### 3.3. Stereo Algorithm

The performance of a matching cost can depend on the algorithm that uses the cost. Thus, we consider three different stereo algorithms: a window-based method, the semi-global method, and a graph-cut method. We implemented each of the matching costs for each stereo method, except for NCC, ZSAD, and ZNCC, which can only be used with the window-based method.

#### 3.3.1. Window-Based

The window-based method [9] aggregates the matching cost over the window. Then, the disparity with the lowest aggregated cost is selected. After selecting the disparity, a subpixel interpolation, followed by a left–right consistency check, is performed in order to invalidate disparity segments and occlusions. The invalid disparity areas are then filled with new disparity values propagated from neighboring pixels. These post-processing steps are used to reduce errors.

#### 3.3.2. Graph Cut

Global methods aim to find correspondences for all pixels in the image via minimizing a global cost function. The graph-cut method [11] is used as one of the global methods for stereo matching. Before performing the graph-cut, a specialized graph is constructed for the energy function to minimize the global cost. The graph-cut method performs a max-flow algorithm to find the minimum cut that minimizes the energy in the specialized graph. The energy function E(D) used in this paper is defined as:(11)$$\begin{array}{c} E\left(D\right)=\sum_{p}\left(C\left(p,D_{p}\right)+\sum_{q\in N_{p}}P_{1}T\left[\left|D_{p}-D_{q}\right|=1\right]+P_{2}T\left[\left|D_{p}-D_{q}\right|>1\right]\right) \end{array}$$
where $C(p,D_{p})$ is a pixel-wise matching cost for all pixel p at their disparities $D_{p}$; *P* coefficients are penalty costs. Either the penalty cost $P_{1}$ or the penalty cost $P_{2}$ is chosen, depending on the disparity difference between the center pixel p and its neighbor q. If the disparity difference equals to one, $P_{1}$ is selected. If the disparity difference is greater than one, $P_{2}$ is selected; therefore, $P_{2}$ gives a greater penalty cost than $P_{1}$.

#### 3.3.3. Semi-Global Matching

The semi-global matching (SGM) [10] algorithm solves the 2D energy function E(D) by reducing it to a large number of 1D problems. The energy function E(D) used in SGM is the same as that of the graph-cut. Unlike the graph-cut, which globally computes the energy function E(D), SGM computes E(D) along the eight 1D paths toward the target pixel. The costs computed along the paths are summed. The summed result is the cost of the pixel of interest. The cost is computed for all disparity levels, and the disparity with the lowest cost is selected. After choosing the disparity, the post-processing steps that were applied to the window-based method are also applied to SGM.

### 3.4. Summary

To thoroughly evaluate the matching performance according to the image filter and matching algorithm, we have tested all possible combinations of filters and stereo matching costs with each stereo algorithm described below section. The combinations we tested in the experiments below are: AD, BT, ZSAD, NCC, ZNCC, Mean/AD, Mean/BT, Mean/ZSAD, Mean/NCC, Mean/ZNCC, LoG/AD, LoG/BT, LoG/ZSAD, LoG/NCC, LoG/ZNCC, BilSub/AD, BilSub/BT, BilSub/ZSAD, BilSub/NCC, BilSub/ZNCC, RankAD, Rank/BT, Rank/ZSAD, Rank/NCC, Rank/ZNCC, and Census. While all filters can be used regardless of the matching algorithm, window-based matching costs (SAD, ZSAD, NCC, and ZNCC) cannot be implemented with the semi-global matching (SGM) or the graph-cut (GC) methods. This is because SGM optimizes the disparity values along 1D paths in eight directions, and the GC method optimizes the disparity map error globally. Because they are optimizing the disparity value globally or in 1D paths, it is not suitable to use the window-based matching cost. Thus, due to implementation differences, window-based matching costs cannot be applied to semi-global and GC methods. On the other hand, the window-based algorithm can aggregate all possible combinations of matching costs and filters.

## 4. Experiments

We generated synthetic active stereo images, which can control parameters affecting the attribute of IR images, from a public passive stereo dataset using an image processing tool [28] to quantitatively evaluate active stereo techniques against the ground-truths. The pattern texture in the synthetic dataset imitates that of an active stereo camera to confirm that our evaluation is valid in a real-world environment qualitatively. Figure 2 shows left images of some sets that synthetically generated an active stereo dataset from the Middlebury 2014 [27] stereo dataset.

We conduct the quantitative and qualitative evaluation by changing four attributes of active IR images on a synthetic active stereo dataset. The first parameter is the *pattern intensity*, which refers to the intensity of the pattern dots emitted from the IR projector. The second parameter is the *pattern contrast*, which refers to the relative brightness difference between the intensity of pattern dots and the intensity excluding pattern dots in the IR image. The third parameter is the *number of pattern dots*, and in this experiment, the density of pattern dots is controlled. Finally, we change the gain of the input IR image and the overall brightness of the active stereo image. We analyze how changes in these parameters affect the accuracy of disparity estimation in our experiments. Figure 3 shows the input left images according to the change of each parameter.

We also analyze changes in these parameters on real IR images captured using commercial RGB-D cameras, RealSense D455. Four attributes of the active IR images on a synthetic active stereo dataset correspond to the laser power, the illumination ratio of the laser power and the external light source, the number of IR projectors, and the illumination of the external light source, respectively. These attributes are adjustable. By comparing the trend from the experiment on the real active dataset with the experiment on the synthetic active stereo dataset, we analyze how the four parameters affect the real IR images.

### 4.1. Dataset and Implementation Details

***Synthetic Active Stereo Image Generation.*** Generally, the active pattern projected from the laser-based IR projector that follows the inverse square law is designed to emit random dots. Moreover, the laser speckle must appear on the object’s surface. To fully replicate the active pattern produced by the widely used off-the-shelf RGB-D camera (i.e., RealSense D455 [29]) to our synthetic dataset, we empirically measured the size of an IR dot, which is a composite of the active pattern. We measured the size of the dot by projecting the pattern onto the fit plane from 1 m away. From this environment, we discovered that the radius of each dot is 5 mm. First, we randomly sampled the location of each pattern dot using Kocis et al.’s method [30]. After randomly sampling the location of each dot, we applied a 2D Gaussian kernel to each dot to generate its speckle. Then, we applied the inverse square law to adjust the size and intensity of each dot based on depth. We fine-tuned the parameter for the 2D Gaussian kernel and the inverse square law so that the size of a dot was 5 mm when it was 1 m away. Then, the RGB image was converted to a gray-scale image, assuming that the receiving wavelength of the IR sensor was encoded with the intensity of the monochrome imaging sensor converted in the RGB camera. Finally, the gray-scale image and the image projected with random patterns were integrated to generate a synthetic active stereo image. We used the standard Middlebury 2014 stereo datasets (*Adirondack, Backpack, Bicycle1, Classroom1, Motorcycle, Piano, Pipes, Playroom, Playtable, Recycle, Shelves, Sticks, Sword1, Sword2*, and *Umbrella*) to generate a synthetic active stereo image [27].

***Implementation Details.*** The disparity range was set to pre-defined ground-truth values for each set. The resolution and disparity ranges of the images were downsampled to one-third of their original size. We heuristically tuned and set the default values of the four attributes of active IR images for all stereo matching algorithms. After setting the default value, only each parameter was changed in the experiment on four attributes of the active IR images. The *pattern intensity* had 60 levels, while the *pattern contrast, number of pattern dots*, and *global gain* of the input images had 20 levels. In the preprocessing filter, the size of all image filters was 7×7. The parameter of the LoG filter (Equation 2) was σ=1, and the parameters of the BilSub filter (Equation 3) were $\sigma_{s}=5$ and $\sigma_{r}=100$, respectively. For the window-based method, we set the window size of all combinations to 9×9 and the aggregation filter size to 15. The smoothness parameters of the SGM method, namely, $P_{1}$ and $P_{2}$, were the same as the parameters used in [13] ($P_{1}$ refers to small disparity differences of the neighboring pixels, and $P_{2}$ was adapted to the local intensity gradient by the neighboring pixels, respectively). The smoothness parameter of SGM was set as *value × number of image channel × matched block size × matched block size* in the StereoSGBM library of *OpenCV* $\left(P_{1}=8\times 1\times 3\times 3,P_{2}=32\times 1\times 3\times 3\right)$ [13,31]. The smoothness parameters of GC were the same as the parameters used in [11] ($P_{1}$ is the same as the GC parameter $P_{1}$ and $P_{2}$ is used to double the value of four gradients below a given threshold, respectively). For the smoothness parameter of GC, the max smooth value was 1 and the weight of the smoothness term was 8000 in the MRF library [32], respectively. The filter size of Census and rank was 9×9. Figure 4 illustrates the input images to which image filtering was applied.

We computed the proportion of pixels with a pixel difference greater than one between the ground-truth disparity value and the estimated disparity image for quantitative evaluation. In the calculated proportion, we cannot determine the disparities in the area where occlusion occurs by matching stereo images. Therefore, these occluded areas are ignored in the proportion calculation.

The software for the window-based and SGM methods used in this paper was implemented using *C++* with *OpenCV* [31]. For the graph-cut method, we experimented with the GC stereo algorithm using the MRF library provided by [32], an open library that has already been implemented. We used a desktop machine equipped with an *Intel Core i7 CPU* and a single GPU of *Nvidia Geforce GTX 2080 Ti* for our experiments.

### 4.2. Evaluation of Synthetic Active Dataset

In this section, each combination is tested on the default setting of synthetically generated active stereo images. Figure 5 shows the performance of each combination in terms of an average error of disparity maps generated from our Middlebury dataset [27] classes.

Figure 5 shows the errors produced by matching costs when aggregated with the window-based method. Using the window-based cost aggregation method, Census produced the lowest average error, and the mean filter with NCC produced the highest average error. The mean filter produced more errors than the other filters because it removed low-texture areas from the image, making it more difficult for the window-based algorithm to distinguish between the differences in the windows. As a result, the probability of false matches increased. The performance of SAD, BT, ZSAD, and NCC varied according to the filter used, but the performance of ZNCC did not. ZNCC is robust to intensity contrast and noise through normalization, and compensates for intensity offsets between left and right images. These advantages allowed ZNCC to show steady performance, regardless of the filters applied to it. When comparing the filters except for Census, the BilSub filter produced the lowest error on average. The BilSub filter locally removed the background that might interfere with matching.

Figure 5b shows the errors produced by matching costs when aggregated with the semi-global matching method. Similar to the result in Figure 5a, Census produced the fewest errors. The most errors were produced by the BilSub filter with BT, which is unlike the window-based method. This vividly shows that BT produces higher errors than AD when it is applied with or without filters. A similar effect of BT can be observed in Figure 5a. BT is known for its robustness against the sampling effect by linearly interpolating the intensity with surrounding neighbors. Because the patterns are projected in the IR environment, patterns do not interfere with the natural intensity of an object. Therefore, the use of linear interpolation causes an inaccurate representation of intensity around the borders of the projected patterns. This blur effect makes it difficult for the stereo algorithm to find a match. On the other hand, AD computes costs by simply subtracting two images at a certain disparity, Thus, AD is able to obtain a more accurate disparity map than BT in an active stereo environment.

Figure 5c shows the errors produced by matching costs when aggregated with the graph-cut method. Census shows the best performance, and the rank filter with AD shows the worst performance. GC seems to produce results that are very different from the window-based and SGM methods. We observed that BT produced more errors than AD, and that the rank filter performed relatively better than some matching costs (Figure 5a,c). However, Figure 5c shows that BT performed better than the AD, and that the rank filter showed the worst performance.

Figure 6 shows the qualitative results of all matching costs tested under the default settings of the attribute. In general, the matching costs applied with the window-based stereo algorithm produced disparity maps similar to the ground-truth. While the matching costs aggregated by SGM showed finer details of the objects’ boundaries, the disparity map generated by GC displays some regions that were filled with the supposedly occluded disparity values. This inaccurate filling of holes causes the misinterpretation of the disparity maps. Overall, Census seems to produce the most accurate result, and AD, with the mean filter applied onto the images, produces the least accurate representation, due to the blurring effect near the boundaries of the objects.

### 4.3. Evaluation on Pattern Intensity Changes

We evaluated each combination at 60 levels (from 5 to 300) that specify the magnitude of the 2D Gaussian kernel. The patterned texture of our synthetic active stereo image was generated using a 2D Gaussian kernel that mimicked a laser speckle. Following the inverse square law, the intensity of each pattern dot depends on the distance between the surface and the camera, which is the depth. The further the distance to the surface, the lower the intensity of the pattern is set. Even though pattern dots, which are close to the camera, have already reached the intensity of 255, pattern dots on the far surface still have room to increase. Thus, we tried to increase the intensity of the pattern dots far from the camera. Figure 7 and Figure 8 show the errors generated by the window-based, SGM, and GC methods when the *pattern intensity* was varied. To begin with, matching costs are grouped based on the filters applied to the image in Figure 7. The performance of the matching costs is positively correlated with the *pattern intensity*, regardless of whether filters are used. When comparing against other non-filtered matching costs, Census performed the best. When the *pattern intensity* was below a certain level, BT performed the worst, but above that level, NCC performed the worst. The performances of ZSAD, NCC, ZNCC, and Census are relatively robust to changes in the *pattern intensity* because normalization and zero-mean subtraction amplify the projected patterns with low intensity. Similar to NCC, ZSAD performed better than SAD when the *pattern intensity* was set higher than 110, and vice versa. The robustness of ZSAD’s, NCC’s, ZNCC’s, and Census’s performances are not lost even when filters were applied because the filters already applied these matching costs’ effects. However, the errors produced by these matching costs quickly saturate when applied with filters. Thus, the graphs in Figure 7b–e show an exponential increase in errors when the *pattern intensity* was extremely low. Census performed the best for the mean, LoG, and rank filter. The stereo matching results produced using the BilSub filter are unclear because all matching costs except for BT produced a similar amount of errors. The results produced from the images filtered by the BilSub filter cannot be clearly stated because all matching costs except BT produced a similar amount of errors. The worst-performing matching cost was NCC for the rank and the mean filter, and for the BilSub filter, the worst-performing matching cost was BT. The worst-performing matching cost for the LoG filter depends on the *pattern intensity*. When the *pattern intensity* is lower than 135, BT performs worst. When the *pattern intensity* is higher than 135, ZSAD performs worst. As a result, the errors exponentially increase when the *pattern intensity* drops below a certain level.

Figure 8a illustrates the errors generated by matching costs in SGM at each level of the *pattern intensity*. From Figure 8a, we can observe that Census is the most robust matching cost, and that it is the best-performing matching cost. In contrast to Census, AD with the LoG filter is the least-robust matching cost against changes in the *pattern intensity*. When the *pattern intensity* decreases below 25, AD with the LoG filter’s error increases exponentially and performs the worst. When the *pattern intensity* is higher than 25, the BilSub filter with AD produces the highest error, and the rate of change in errors is not stable, unlike other matching costs. The error produced by the BilSub filter with AD heavily fluctuates relative to other matching costs. This is because the performance of the BilSub filter is strongly dependent on the scene structure of the images. As a result, the images with complex scene structures produce more errors than other images, leading to the fluctuation in errors produced by the BilSub filter with AD.

Similar to the window-based and SGM methods, Census performed the best when aggregation was conducted by GC, as shown in Figure 8b,c. GC produces robust results when matching costs are computed on the images filtered by the BilSub filter and the rank filter. The mean filter applied to images produces fluctuating errors when the *pattern intensity* is low. Since the mean filter computes the mean intensity of images, the information required for matching at low intensity becomes ambiguous in both the pattern texture and the passive texture. This effect becomes dependent on the objects in the scene, causing the average error to be non-constant. Nevertheless, its error is reduced to the same level as other filters when the pattern power is high enough.

The qualitative results shown in Figure 9, Figure 10 and Figure 11 are results produced by the window-based, SGM, and GC methods, respectively. The results from the low (a)–(f) and high (g)–(l) *pattern intensity* are illustrated to qualitatively evaluate the effect of the *pattern intensity* and the performance of the matching costs against changes in *the number of pattern dots*. Overall, regardless of the matching costs used, all stereo algorithms perform poorly when the *pattern intensity* setting is low. Because the pattern dots become indistinguishable from the background texture, mismatches and inaccurate estimations of disparity values for each pixel are caused. When the *pattern intensity* is set sufficiently high, the pattern dots become distinguishable, and regions are filled with complex pattern textures. Thus, holes created by matching errors are filled with correct disparity values, and the results show the accurate contour of scene structures. The matching cost that produced the least difference between the two settings is Census.

In summary, all matching costs’ performances are positively correlated with the *pattern intensity*. However, their performance saturates from a certain level of the *pattern intensity*. The performances of the matching costs are almost even with each other when using the window-based algorithm with the BilSub filter applied to the images. This is because the BilSub filter strengthens the edge features while killing the planar features. Because the pattern dots are small, the BilSub filter heavily emphasizes the pattern texture in the image. As a result, only the pattern texture is used for matching as the *pattern intensity* increases. Thus, the performance of all matching costs, except BT, produces similar errors when using the BilSub filter with the window-based method. SGM produces variable numbers of errors when using the AD and BilSub filters because SGM depends on both pattern and passive textures for matching. The number of mismatches produced by SGM was positively correlated with scene complexity because it solely relied on the pattern texture and optimized the disparity value along the eight lines running toward the target pixels. Thus, SGM becomes biased toward each pixel and, thereby, produces errors in images with complex scene structures. In order to investigate the effect of the correlation between object and pattern textures, we evaluated how matching cost performance varied by the *pattern contrast*.

### 4.4. Evaluation of Pattern Contrast Changes

After discovering the saturation of the errors produced by matching costs, we tested the matching costs on the change in the *pattern contrast*, which defines the inverse relationship between the passive texture and pattern texture. The *pattern contrast* is a significant factor that influences the performance of matching costs and stereo algorithms, as shown in Figure 12 and Figure 13. The *pattern contrast* is the intensity ratio between the IR image and the pattern image. The *pattern contrast* is set in the last stage of generating a synthetic active stereo image. The x-axis in Figure 12 and Figure 13 refers to 20 levels (from 0.05 to 0.1) of the intensity of the pattern image, relative to the intensity of the IR image. For example, if the relative intensity of the pattern image is set as 0.05, the relative intensity of the IR image is set as 0.95. We analyzed how the passive texture, which is the original IR image excluding the pattern dots, and the pattern texture, which is the pattern dots projected onto the image, affect each other on stereo matching. To analyze the disparity estimation accuracy for the passive texture and pattern texture intensity ratio, we measured the error by changing the intensity ratio of the pattern image and the original IR image, excluding the pattern dots. Figure 12 and Figure 13 show the quantitative results of the *pattern contrast* change experiments. This shows that the disparity estimation is inaccurate not only when the intensity of the pattern texture is low, but also when the passive texture is too dark. Therefore, it is necessary to find the optimal *pattern contrast* level for accurate disparity estimations.

Figure 12 groups the results by the filters applied to the active stereo images in the window-based method. Overall, all of the results show a parabolic shape. However, except for ZNCC and Census in Figure 12a, Figure 12a still shows tendency in Section 4.3. The results from the BilSub and rank filter show a more definitive parabolic shape, which means more errors are produced when the contrast is set at an extreme level. Because the BilSub and rank filters remove weak features from the image when the contrast is low, the pattern texture is erased, and when the contrast is high, the passive texture is erased.

The errors produced by matching costs with SGM are shown in Figure 13a. Census produces the fewest error throughout the contrast level range. The results of the filters, except for Mean/AD and BilSub/AD, are similar to those from Figure 12. The performances of Mean/AD and BilSub/AD are notable. The BilSub filter removes the passive texture, and the mean filter blurs the passive texture from the images. Therefore, matching is performed by using pattern texture only. For these reasons, in the images with complex scene structures, many similar pattern dots may cause mismatches by the SGM method. As a result, these filters produced many errors from images with complex scene structures, and a few from simple ones.

Similar to the result from the SGM method, Figure 13b,c illustrates that the best-performing and the most-robust matching cost is Census, when used with GC as the stereo algorithm. The least-robust matching costs, when used with GC, are AD, Mean/AD, BT, and Mean/BT. These matching costs do not remove noise and unnecessary information from the images. Thus, when globally optimizing the disparity map, these offsets negatively affect the performance of the active stereo technique.

In addition, when the *pattern contrast* is low, the pattern texture is barely visible in the image. As a result, the performance of BT and AD is similar to that of the passive stereo technique. Furthermore, the mean filter blurs boundaries between the passive texture and the pattern texture. Hence, the images are contaminated when the *pattern contrast* is low. However, as the *pattern contrast* increases, it becomes obvious for GC to match correct pixels because the feature of pattern texture becomes distinctive.

Figure 14, Figure 15 and Figure 16 illustrate the qualitative results of the matching costs obtained from the window-based, SGM, and GC methods, respectively. For all stereo algorithms, the performance of the matching costs improved with the increase in the *pattern contrast*. However, when comparing with the results from the default settings in Figure 6, we observed that the results of BilSub/SAD and Rank/SAD are better with the default settings. Nevertheless, the qualitative result of Census produced by GC showed almost uniform results and generated a disparity map similar to the ground-truth of a scene structure.

In summary, finding the optimal *pattern contrast* level is crucial for the matching costs. In the window-based method, Census and ZNCC also show that these optimal levels should be found while showing the least overall errors and robustness. Because Census depends on the relative ordering of image intensity, matching corresponding points invariant to monotonic variations of illumination is possible. Changing the level of contrast can change this relative ordering. As shown in Figure 12a, the lowest error appears at the appropriate contrast level. For the ZNCC cost, only the intensity change in the image kernel is used for matching the corresponding point. Therefore, the ZNCC cost also shows a tendency similar to that of Census. On the other hand, changing the contrast level in the non-filter only scales the image intensity value. The window-based method, using the image intensity value, dramatically increases or decreases the cost value, depending on the scale. Therefore, SAD, BT, ZSAD, and NCC show the trend shown in Section 4.3. We have evaluated the effect of the *pattern intensity* and *pattern contrast* on the performance of matching costs. These parameters affect the feature strength of the pattern texture. However, we have not evaluated the effect of the *number of pattern dots* filling the scene. Thus, in the next section, we evaluate the performance of matching costs on changes in the *number of pattern dots*.

### 4.5. Evaluation of the Number of Pattern Dots Changes

While the *pattern intensity* and *pattern contrast* control the extent of patterns’ definition in the images, the *number of pattern dots* determines the complexity of the pattern texture applied to the images. Thus, we evaluated the performance of matching costs against changes in the *number of pattern dots*, and the results are shown in Figure 17 and Figure 18.

The results of changing the *number of pattern dots* on each matching with the window-based stereo algorithms are shown in Figure 17. As the *number of pattern dots* increases, so does the matching cost performance, but this eventually increases plateaus. The performance improves because more textures are applied to less strongly textured regions in the images, leading to fewer false matches.

Figure 18a shows that the matching cost performance with the SGM cost performance was positively correlated with the *number of pattern dots*, except for that of the Census method, which was relatively unchanged. However, the Census method produced the lowest number of errors. The Census method performed the best and was robust against changes in the *number of pattern dots* when using SGM as the stereo algorithm. The performance of BilSub/AD does not show a smooth curve, unlike other matching costs, because the BilSub filter does not perform well with certain images having complex scene structures. As a result, the performance of the BilSub filter fluctuates more than other matching costs. In addition to results from window-based and SGM methods, the performance of matching costs aggregated by GC are shown in Figure 18b,c. Unlike the results of SGM, the LoG and BilSub filters do not show a tendency to converge as the *number of pattern dots* increases.

The qualitative results shown in Figure 19, Figure 20 and Figure 21 are results produced by the window-based, SGM, and GC methods, respectively. The results from the low (a)–(f) and high (g)–(l) numbers of pattern dots are used to qualitatively evaluate the effect of the *number of pattern dots* on the matching cost performance. Overall, regardless of the matching costs used, all stereo algorithms performed poorly when the number of the pattern dots was low. This is because the inaccurate disparity values are estimated from the textureless regions in the images, leading to a mismatch of pixels in the left and right images. Even though all matching costs performed poorly, we observed that Census could produce a disparity map that is relatively similar to the ground-truth. When the *number of pattern dots* increases, the textureless regions are filled with complex pattern textures. Thus, holes created by matching errors are filled with correct disparity values, and show the accurate contour of scene structures.

To summarize, the *number of pattern dots* strongly affects the performance of matching costs. As expected, in an active stereo environment, the pattern gives additional information to matching, so the more pattern dots in the local method, the better the performance and the tendency to saturate. On the other hand, mean, LoG, and BilSub, which are smooth filters, do not converge in the global method but fluctuate significantly. Like the fluctuation observed in Section 4.4, the smooth effect reduces the features of the passive texture for matching and increases the feature of the pattern dots. The BilSub filter has a strong smoothing effect because it subtracts the part, except for the edge, from the original image. The mean and LoG filters have a weak smooth effect because they blur the image. This strong smoothing effect makes it impossible to match the features of the passive texture in both SGM and GC, causing a fluctuating error. A weak smooth effect weakens the features of the passive texture in both SGM and GC. However, some information remains, causing a convergence of errors. Moreover, inferred from these analyses, the experimental results of SGM and GC for LoG have a smooth intermediate effect. The *pattern intensity, pattern contrast*, and *number of pattern dots* are attributes of the projector affecting the IR images. Not only the attributes of the projector, but also external attribute that affects the IR images, should be analyzed. Therefore, in the next section, we tested the active stereo techniques against changes in the *global gain* to find the effect of global brightness on the active stereo technique.

### 4.6. Evaluation of Gain Changes

In this section, we evaluated the performance of the matching costs and stereo algorithms on the changes in the *global gain* of the images. The results produced by matching costs aggregated by the window-based, SGM, and GC methods are illustrated in Figure 22 and Figure 23, respectively. As shown in Figure 22, the matching costs aggregated by the window-based method are very robust against changes in the *global gain*. Because the increase and the decrease in the *global gain* cause the intensity of background texture and pattern texture to change linearly, the performance of the matching costs is not affected by this change. While the images that are or are not filtered by the mean, LoG, and rank filters do not affect the performance of matching costs, the BilSub filter causes the performance of the matching costs to fluctuate. This fluctuation is caused by the subtraction of background texture by the BilSub filter. The accuracy of the background subtraction by the BilSub filter depends on the window size. Because we applied the BilSub filter with the same window size for all images, the BilSub filter may not perform well for some images with complex scene structures. Thus, the performance of the matching costs on the images filtered by the BilSub filter severely fluctuates. For the window-based method, Census produced the fewest errors for all *global gain* levels. The worst-performing matching costs differed based on the filters applied to the images. NCC performed worst when used on non-filtered images and on images filtered by the mean and rank filters. However, for images filtered by the LoG and BilSub filters, BT produced the most errors. We did not include the tables, because we could not determine the optimal settings for the attribute pattern producing the fewest errors.

When SGM is used as the stereo algorithm, the matching costs show similar behavior to the window-based method, as shown in Figure 23a. Census produces the fewest errors, and BilSub/AD produces the most errors. In addition, the error of the BilSub filter fluctuates the most, compared to other filters. The result of BilSub/AD shows a drastic increase in errors produced when the *global gain* exceeds a certain level. The subtraction of background texture leaves the image with a pattern texture only. Due to the limitation of the data size of each pixel, the intensity of the pattern cannot exceed 255. As the *global gain* increases, the intensity of the patterns reaches the maximum level. As a result, the difference between the pattern dots representing different depths cannot be distinguished. Due to this issue, SGM with the BilSub filter produces more errors as the *global gain* reaches a certain level.

While the SGM and window-based methods show that the errors either saturate or decrease with the increase in the *global gain* for most matching cost, GC shows that some matching costs, especially BT and AD, produce more errors as the *global gain* increases. As mentioned in the previous paragraph, some distinction between patterns is lost due to the pixel reaching its maximum intensity. Because GC aggregates costs in a 2D perspective, it is more likely for GC to make a mismatch and compute an erroneous disparity. As a result, the increase in the *global gain* causes more data to be lost globally. Overall, more errors are produced. However, Census shows the high robustness against *global gain* changes even when aggregated by GC, and still performs better than other matching costs. On the other hand, BT and AD perform worse as the *global gain* increases.

The qualitative results shown in Figure 24, Figure 25 and Figure 26 were produced by the window-based, SGM, and GC methods, respectively. The figures from (a) to (f) are the results obtained at the low *global gain*, and those from (g) to (l) are the results obtained at the high *global gain*. The qualitative results of the window-based and SGM methods follow a trend similar to the quantitative results. As explained in the quantitative results, there are no significant differences between the low and high *global gain* results. On the other hand, the results from GC show that more details are captured from the images when the *global gain* increases. For example, there is a large blob around the chair, and a disconnection of the chair’s arm in Figure 26b. This inaccurate representation is caused by the blurring effect of the mean filter on the images. However, the finer details of the chair are represented with an accurate disparity value when the *global gain* was set high.

In summary, changes in the *global gain* did not affect the performance of matching costs when used with the window-based method and SGM. However, AD and BT showed a subtle increase in errors produced with the *global gain* when used with GC because the likelihood of making mismatches increases due to the smoothness term used in GC. Overall, Census produces the least errors and is robust against changes in the *pattern contrast*, regardless of the stereo algorithm used. The worst-performing matching costs for the window-based method are NCC and BT, and for SGM it is NCC. For GC, BT and AD applied on images filtered with the rank filter perform worse when the *global gain* is low. When the *global gain* surpasses a certain level, BT and AD without filters produces the most errors.

### 4.7. Comparison of Runtime

We measured the runtime of all the combinations tested in our experiment, and Table 2 shows the measured runtime of each combination. The results shown in Table 2 are measured under default settings of the pattern attributes. When comparing the runtime in terms of the matching algorithm, the window-based algorithm is faster than SGM and GC. The high runtime of the global method is caused by the optimization process of iteratively reducing the error.

A comparison of the runtime of the matching costs for each algorithm helped us to conclude that AD and SAD took the least amount of time and that Census took the longest time to finish their respective tasks. AD and SAD take the least time because they compute matching costs by simply subtracting the target and source intensity values. NCC takes longer than SAD and BT because it takes the extra step of normalizing the values within the window and performing cross-correlation. For ZSAD and ZNCC, they take a longer time than their original forms, which are SAD and NCC, respectively. This is because the values within the window are first processed to make the mean equal to zero, and then the matching cost is computed. Census takes the longest time because it first requires the binary encoding of values within the window to determined its intensity ordering.

Due to a high runtime, the global method is not applicable to dynamic object reconstruction, but is applicable to high-quality static scene reconstruction. On the other hand, the window-based method is more suitable for real-time applications, such as dynamic object reconstruction, even though it produces less accurate results than the global methods.

## 5. Evaluation of the Real Active Images

### 5.1. Quantitative Results

We used the root mean square (RMS) error of Keselman, Leonid, et al. [24] to obtain quantitative results on real active images:(12)$$\begin{array}{c} \epsilon_{\%ofz}=\frac{\epsilon_{mm}}{z}=100\times \frac{1}{z}\sqrt{\sum_{i=1}^{n}\frac{{\left(z_{i}-z\right)}^{2}}{n}} \end{array}$$
where *z* is the best-fit plane value that is provided by RealSense SDK [34], $z_{i}$ is the estimated depth value of the *i*-th pixel, and *n* is the number of pixels of the plane position in the images. We aligned a flat plane perpendicular to the off-the-shelf RGB-D camera (i.e., RealSense D455 [29]). To express four pattern attributes that affect matching cost performance, we used factors of RealSense SDK and external environments, such as the laser power, the illumination ratio of the laser power and the external light source, the number of IR projectors, and the illumination of the external light source. Table 3 shows the quantitative results obtained through experiments described in below sections. The overall quantitative results for the real active images follow a similar trend to the results of those for the synthetic active images.

Census most-accurately estimates the disparity values for all stereo matching methods. The least-accurate matching cost differs based on the filter and stereo algorithm used. When comparing the results between the stereo matching methods, GC produces less errors than the others.

### 5.2. Qualitative Results

After fully evaluating the active stereo matching algorithms using the synthetic dataset, we conducted the same experiments using a real dataset that was captured using an Intel RealSense D455 camera that contained stereo IR cameras and an IR pattern projector. The attributes of the active IR images can be controlled to simulate the same settings used for the evaluation of the synthetic dataset, as shown in Figure 27. While the *pattern intensity* and *global gain* can be adjusted by a single device, simulating the changes in the *number of pattern dots* and *pattern contrast* required a more complex solution. Because each IR pattern projector projects the specific number of patterns, we utilized nine devices of the same model and positioned them to project patterns toward the same field of view. We simulated the change in the *number of pattern dots* by changing the number of devices used accordingly. The adjustment of *pattern contrast* was simulated by controlling the brightness of the external light source and the *pattern intensity*. When obtaining an image with the high *pattern contrast*, we set the brightness of the external light source low and the *pattern intensity* high, and vice versa.

***Default Settings.*** The results shown in Figure 28 are obtained under the default conditions. When comparing the results based on the performance of the matching costs, Census produced the fewest holes and mismatches. The results of AD, Mean/AD, LoG/AD, and BilSub/AD show a disconnection in a chair leg, while Census and Rank/AD captured the entire structure of the chair. The distinct difference between Census and Rank/AD can be found in the holes generated on a computer monitor. There are bigger holes in the Rank/AD than in Census on the monitor, representing more errors generated by Rank/AD.

***Pattern Intensity Changes.*** The results in Figure 29 are obtained by changing the *pattern intensity*. The top and bottom rows of the results for each stereo algorithm show the results produced with low and high pattern intensities, respectively. In terms of changes in the *pattern intensity*, the performance of all matching costs and the stereo algorithms improve when the *pattern intensity* is set high. Because the pattern texture becomes more distinguishable, features of pattern texture become more distinct. Thus, the holes in the results from the low-pattern-intensity images are correctly filled in the results from the high-pattern-intensity images. When comparing the results between the matching costs, Census performed better under both low- and high-pattern-intensity settings than other matching costs. Under low-pattern-intensity conditions, Census could yield the correct representation of the chair and mannequin. In contrast, other matching costs show a disconnection in the chair’s frame and inaccurately fill holes between the arms and torso of the mannequin. Under high-pattern-intensity conditions, the qualitative result of Census shows fewer holes than other matching costs.

***Pattern Contrast Changes.*** The results in Figure 30 are obtained by changing the *pattern contrast* and setting the other attributes at default. The top and bottom rows for each stereo algorithm show the results produced with the low and high *pattern contrasts*, respectively. Following a similar trend as the *pattern intensity*, the increase in the *pattern contrast* causes an increase in the performance of the matching costs because the complex pattern texture becomes more vivid. When comparing the results between the matching costs, Census produces the fewest errors in the disparity map. Census performed better than AD, Mean/AD, LoG/AD, and BilSub/AD because Census visualized the structure of the chair, mannequin, and studio more accurately than those other matching costs. Rank/AD performed outstandingly when the window-based method and SGM were applied. However, when GC is used, many errors can be witnessed in the results, while Census maintains its high performance.

***Number of Pattern Dots Changes.*** The results in Figure 31 are obtained by changing the *number of pattern dots* and setting the other attributes at default. The top and bottom rows for each stereo algorithm show the results for low and high numbers of pattern dots, respectively. The effect of the *number of pattern dots* projected onto the scene is visualized for all matching costs. Numerous holes generated due to few pattern dots projected are filled when the *number of pattern dots* increases because the complex texture is added to textureless areas, prohibiting mismatches from occurring. Even with the increase in the *number of pattern dots*, the matching costs could not estimate the disparities of the computer monitor. On the other hand, many errors occur around the chair when SGM is used. Nevertheless, when used with Census, GC could produce an accurate representation of the chair, monitor, mannequin, and studio background under both a low and high number of pattern dots.

***Gain Changes.*** The results in Figure 32 are obtained by changing the *global gain*. The top and bottom rows for each stereo algorithm show the results produced with the low and high *global gain*, respectively. The synthetic dataset evaluation showed that changes in the *global gain* did not affect the result of the active stereo matching. However, when testing the effect of the *global gain* on the real dataset, we discovered some improvements in the results with the increase in the *global gain*. The matching costs did not perform well with textureless regions, such as the computer monitor, and complex structures, such as the chair. With the increase in the *global gain*, the features of these regions become more noticeable and allowed the active stereo matching technique to match the pixels correctly. Overall, Census produced the fewest errors around the chair and computer monitor, proving, thus, that Census is the best-performing matching cost.

## 6. Discussion and Conclusions

The previous surveys [13,14,15] that evaluated passive stereo matching techniques analyzed various attributes, such as radiometric and color differences. While the passive stereo technique relies solely on the information provided by the passive texture, the active stereo technique performs matching using both passive texture and pattern texture. The active stereo matching technique is performed under the IR environment, where the IR pattern is projected onto the scene. The attributes affecting the passive texture under the IR environment are similar to those affecting the passive stereo technique. However, attributes affecting the pattern texture are different. As a result, the results of these previous studies cannot address the full capacity of the active stereo technique. Thus, we analyzed and evaluated the relationship between the attributes affecting the pattern texture and performance of the active stereo technique.

In this paper, we have evaluated how the *pattern intensity, pattern contrast, number of pattern dots*, and *global gain* affect matching cost performance. Through thorough experiments on four pattern attributes (the *pattern intensity, pattern contrast, number of pattern dots*, and *global gain*) that affect the performance of the active stereo technique, we separately analyzed and discussed the results produced by the window-based and global methods.

The window-based method is an algorithm which computes disparity along a scanline using the neighboring pixels of a target pixel. In a general environment of passive stereo matching, textureless regions in an image have similar features; as a result, the ambiguity in matching increases. Consequently, the accuracy of disparity estimations for textureless regions significantly drops. On the other hand, a locally unique pattern is projected onto the image in the active stereo environment. Thus, more accurate matching can be accomplished by comparing the unique features within a window. The results show that as the uniqueness of the pattern features increases, the accuracy of the disparity estimation also increases (Figure 7 and Figure 17).

We evaluated the effect of changing four pattern attributes on the performance of the window-based method. Firstly, as the intensity of the pattern and the number of pattern dots increase, we have discovered that the error in estimating disparity decreases and converges (Figure 7 and Figure 17). At a certain level of the *pattern intensity*, a further increase in the *pattern intensity* ceases to affect the performance of the window-based method. In addition, an extreme increase or decrease in the *pattern contrast* brings out a negative effect on the disparity estimation, due to the loss of the pattern’s uniqueness (Figure 12). From this observation, finding the optimal level of contrast between passive texture and pattern texture becomes a crucial task to produce the least errors in estimations of disparity. In contrast to other pattern attributes, changes in the *global gain* do not influence the accuracy of disparity estimations (Figure 22). The cause of this phenomenon is due to the implementation of the window-based method, because all matching costs compute differences between the windows of the source and the target. Since an increase or decrease in the *global gain* does not change the outcome of the difference between these two patches, the performance of the window-based method stays consistent.

Different from the window-based method, global methods not only compute matching costs at certain pixels, but also reduce disparity inconsistencies between neighboring pixels and discontinuities along the edge via optimization. By optimizing the disparity consistency, global methods can accurately estimate disparity to a certain extent without the unique features given by pattern texture. Due to the optimization process, the runtime of the global methods is higher than that of the window-based method. Even though their runtime is higher than that of the window-based method, global methods show higher accuracy than the window-based method, which simply computes the difference between the source and target windows (Figure 8, Figure 13, Figure 18 and Figure 23).

We performed the same experiment with the global methods as with the window-based method. Similar to the window-based method, as the *pattern intensity* and *number of pattern dots* increase, the error in disparity estimation decreases and converges. However, the error does not dramatically converge, as with the window-basd method (Figure 8 and Figure 18). This is because the global methods produce disparity maps using global optimization, which regularizes pixels with large disparity discontinuity. With regards to the experiment on changes in the *pattern contrast*, the SGM’s result shows a shallower parabolic curve compared to that of the window-based method, while GC’s result does not show any resemblance to either method’s results. Unlike GC, SGM’s result shows a similarity to that of the window-based method because SGM uses dynamic programming to optimize a certain pixel’s disparity values based on other disparity values along 1D paths running in eight directions from the target pixel (Figure 13). Lastly, because global methods perform optimization based on differences in image features, changes in the *global gain* seem to not affect the performance of the global methods (Figure 23).

After evaluating the performance of the window-based and global methods on changes in all attributes, Census seems to show the highest accuracy. Census first sets the value of the reference pixel and its neighbor with a binary number. The neighboring pixels with intensities higher than the reference number are set as 1, and the others are set as 0. Through binary encoding, Census can not only store the intensity ordering but also the spatial structure of the local neighborhood. As a result, Census becomes robust to illumination.

In terms of the accuracy aspect, the global method seems to produce more accurate results than window-based method in estimating the disparity values. However, due to its high runtime caused by the optimization step, the global method is more suitable when applied to high-quality static scene reconstruction. On the other hand, the window-based method is more suitable for real-time applications, such as dynamic object reconstruction, even though it produces less accurate results than the global methods.

The limitation of our experiment on the real dataset was faced when testing the effect of the *pattern contrast* on the real dataset. The *pattern contrast* is a parameter that the IR pattern projector cannot directly control. In order to test the effect of the *pattern contrast* on the real dataset, we controlled the external light source and the projector power to simulate a similar effect of the *pattern contrast*. Indeed, the external light source affects the intensity of pattern dots existing in the IR domain. Thus, in the real dataset, the *pattern contrast* is manually controlled and chosen based on visual feedback. However, the inverse linear relationship between the ratio of the brightness level of the external light source and the *pattern intensity* is kept, and the desired effect is acquired. Therefore, the qualitative result on the *pattern contrast* in the real dataset shows a similar trend as that of the synthetic dataset.

With the results and analysis we obtained from thorough experiments, we assist with the implementation of the active stereo setup. Using these analyses and results, many applications of active stereo matching techniques [35,36,37,38,39,40] will be benefited by using the appropriate combination of matching cost, image filter, and stereo algorithms.

## Figures and Tables

**Figure 1 sensors-22-03332-f001:**
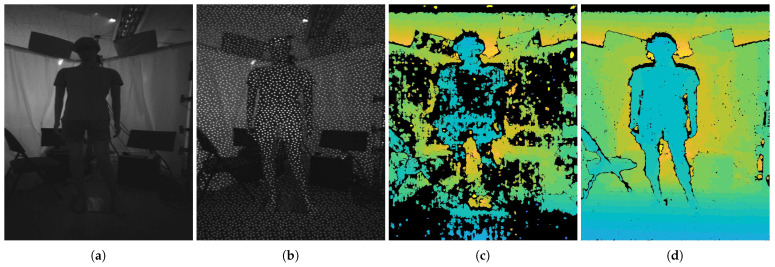
Depth estimation results with and without the active pattern. The active pattern increases the depth estimation accuracy by providing additional information for calculating corresponding points in textureless regions. (**a**) Non-active image. (**b**) Active image. (**c**) Non-active depth result. (**d**) Active depth result.

**Figure 2 sensors-22-03332-f002:**
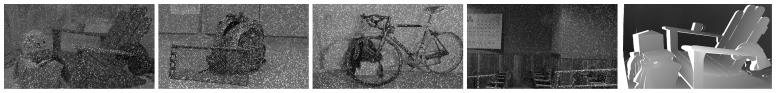
Some sets of left images with synthetically projected pattern. The last image is the GT disparity map of the first image.

**Figure 3 sensors-22-03332-f003:**
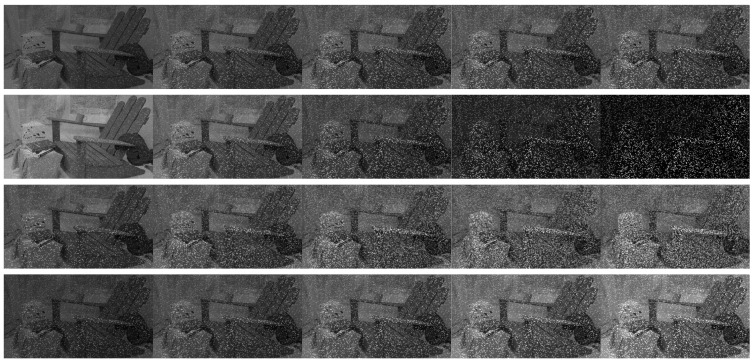
Left synthetic images according to our experimental parameters affecting the attribute of IR images. The figures show changes in the *pattern intensity, pattern contrast, number of pattern dots*, and *global gain* from top to bottom.

**Figure 4 sensors-22-03332-f004:**
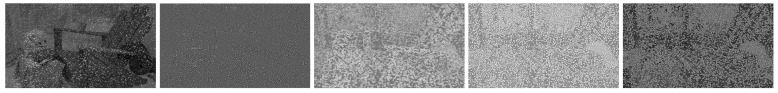
The figures show the filtered images used for combinations for the experiment. From left to right, the figures show images with no filter, the mean filter, the LoG filter, the BilSub filter, and the rank filter applied. For visualization, contrasts of all images have been enhanced through histogram equalizing.

**Figure 5 sensors-22-03332-f005:**
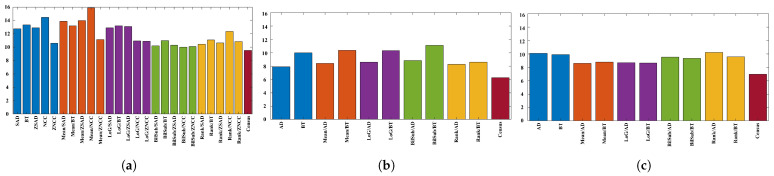
Mean errors on the combinations of the cost and image filters, according to each stereo matching method over the synthetic active dataset. The attributes of the active IR images were set to default values. Each color represents different filters along the methods. (**a**) Window-based. (**b**) SGM. (**c**) GC.

**Figure 6 sensors-22-03332-f006:**
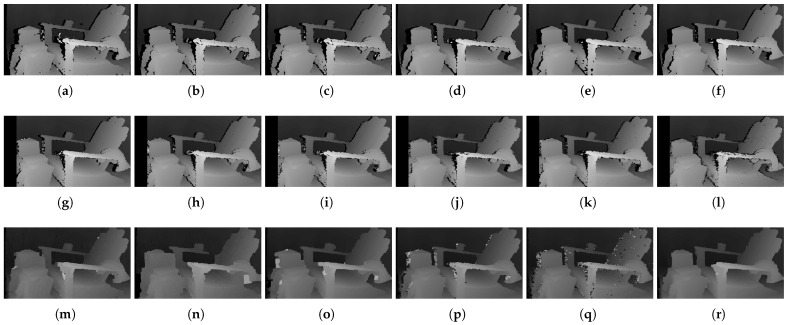
Estimated disparity images of the combinations of the cost and the image filter, according to each stereo matching method over the synthetic active dataset. All attributes of the active IR images were set to default values. In SGM, the columns equal to the max disparity value are not used in the left image because the corresponding pixels in the right image cannot be used for comparison. Therefore, missing pixels appear at the left edge of the image [33]. (**a**) Window, SAD. (**b**) Window, Mean/SAD. (**c**) Window, LoG/SAD. (**d**) Window, BilSub/SAD. (**e**) Window, Rank/SAD. (**f**) Window, Census. (**g**) SGM, AD. (**h**) SGM, Mean/AD. (**i**) SGM, LoG/AD. (**j**) SGM, BilSub/AD. (**k**) SGM, Rank/AD. (**l**) SGM, Census. (**m**) GC, AD. (**n**) GC, Mean/AD. (**o**) GC, LoG/AD. (**p**) GC, BilSub/AD. (**q**) GC, Rank/AD. (**r**) GC, Census.

**Figure 7 sensors-22-03332-f007:**
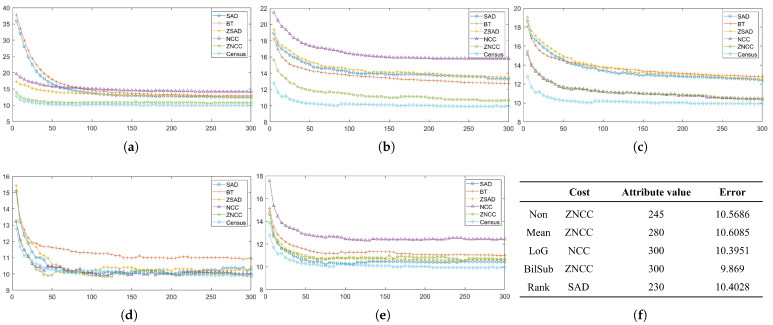
Errors of combinations of the cost and image filter using the window-based stereo method, according to the variation of the *pattern intensity* parameter. (**a**) Non-filter. (**b**) Mean. (**c**) LoG. (**d**) BilSub. (**e**) Rank. (**f**) Best of Window.

**Figure 8 sensors-22-03332-f008:**
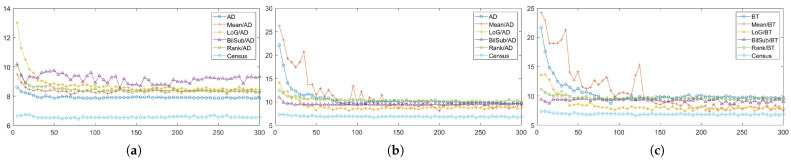
Errors of combinations of cost and image filter using the SGM and GC stereo methods, according to the variation of the *pattern intensity* parameter. (**a**) SGM/AD. (**b**) GC/AD. (**c**) GC/BT.

**Figure 9 sensors-22-03332-f009:**
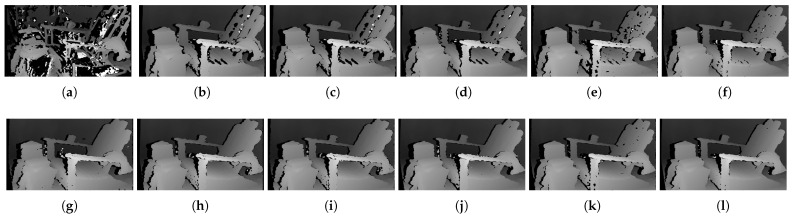
Estimated disparity images of the combinations of cost and image filter using the window-based stereo method, according to the low and high *pattern intensity* parameters. The images in the top row show disparities at the low intensity parameter, and the images in the bottom row show those at the high intensity parameter, respectively. (**a**) Window, SAD. (**b**) Window, Mean/SAD. (**c**) Window, LoG/SAD. (**d**) Window, BilSub/SAD. (**e**) Window, Rank/SAD. (**f**) Window, Census. (**g**) Window, SAD. (**h**) Window, Mean/SAD. (**i**) Window, LoG/SAD. (**j**) Window, BilSub/SAD. (**k**) Window, Rank/SAD. (**l**) Window, Census.

**Figure 10 sensors-22-03332-f010:**
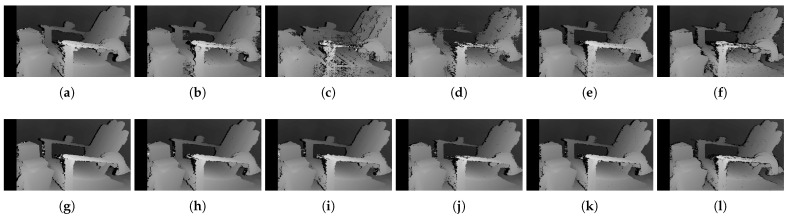
Estimated disparity images of the combinations of cost and image filter using the SGM stereo method according to the low and high *pattern intensity* parameters. The images in the top row show disparities at the low intensity parameter, and the images in the bottom row show those at the high intensity parameter, respectively. (**a**) SGM, AD. (**b**) SGM, Mean/AD. (**c**) SGM, LoG/AD. (**d**) SGM, BilSub/AD. (**e**) SGM, Rank/AD. (**f**) SGM, Census. (**g**) SGM, AD. (**h**) SGM, Mean/AD. (**i**) SGM, LoG/AD. (**j**) SGM, BilSub/AD. (**k**) SGM, Rank/AD. (**l**) SGM, Census.

**Figure 11 sensors-22-03332-f011:**
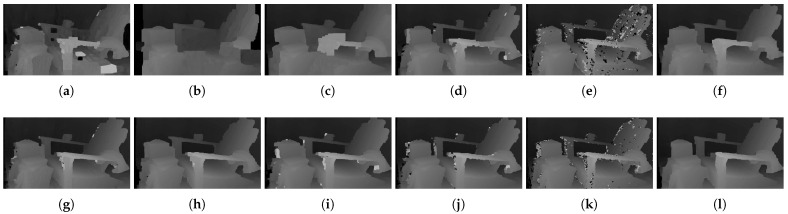
Estimated disparity images of the combinations of cost and image filter using the GC stereo method according to the low and high *pattern intensity* parameters. The images in the top row show disparities at the low intensity parameter, and the images in the bottom row show those at the high intensity parameter, respectively. (**a**) GC, AD. (**b**) GC, Mean/AD. (**c**) GC, LoG/AD. (**d**) GC, BilSub/AD. (**e**) GC, Rank/AD. (**e**) GC, Census. (**g**) GC, AD. (**h**) GC, Mean/AD. (**i**) GC, LoG/AD. (**j**) GC, BilSub/AD. (**k**) GC, Rank/AD. (**l**) GC, Census.

**Figure 12 sensors-22-03332-f012:**
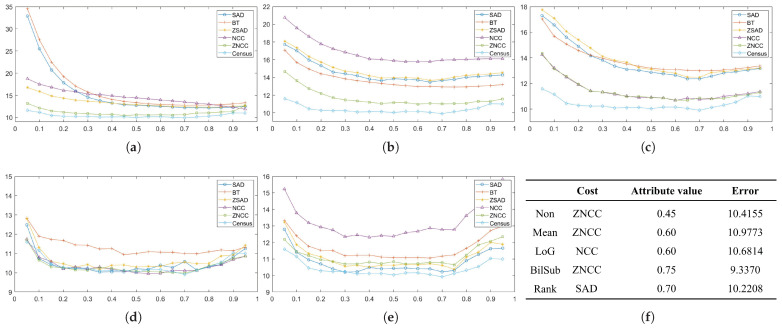
Errors of combinations of cost and image filter using the window-based stereo method, according to the variation of the *pattern contrast* parameter. (**a**) Non-filter. (**b**) Mean. (**c**) LoG. (**d**) BilSub. (**e**) Rank. (**f**) Best of Window.

**Figure 13 sensors-22-03332-f013:**
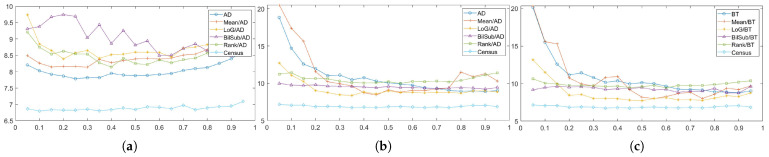
Errors of combinations of cost and image filter using the SGM and GC stereo methods, according to the variation of the *pattern contrast* parameter. (**a**) SGM/AD. (**b**) GC/AD. (**c**) GC/BT.

**Figure 14 sensors-22-03332-f014:**
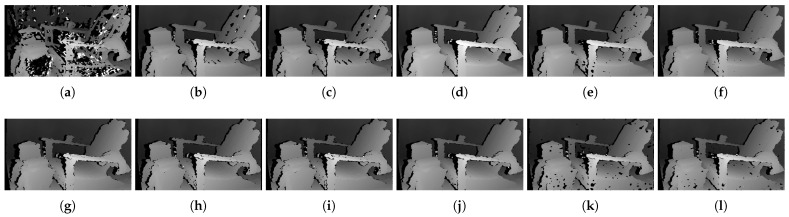
Estimated disparity images of the combinations of cost and image filter using the window-based stereo method, according to the low and high *pattern contrast* parameters. The images in the top row show disparities at low contrast, and the images in the bottom row show those at high contrast, respectively. (**a**) Window, SAD. (**b**) Window, Mean/SAD. (**c**) Window, LoG/SAD. (**d**) Window, BilSub/SAD. (**e**) Window, Rank/SAD. (**f**) Window, Census. (**g**) Window, SAD. (**h**) Window, Mean/SAD. (**i**) Window, LoG/SAD. (**j**) Window, BilSub/SAD. (**k**) Window, Rank/SAD. (**l**) Window, Census.

**Figure 15 sensors-22-03332-f015:**
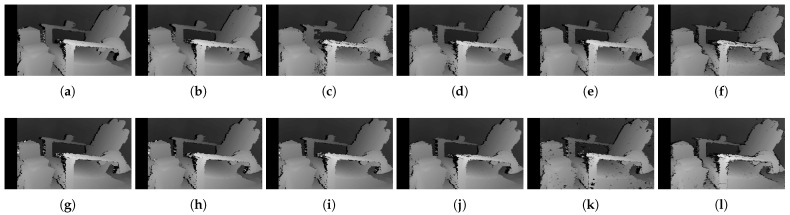
Estimated disparity images of the combinations of cost and image filter using the SGM stereo method, according to the low and high *pattern contrast* parameters. The images in the top row show disparities at low contrast, and the images in the bottom row show those at high contrast, respectively. (**a**) SGM, AD. (**b**) SGM, Mean/AD. (**c**) SGM, LoG/AD. (**d**) SGM, BilSub/AD. (**e**) SGM, Rank/AD. (**f**) SGM, Census. (**g**) SGM, AD. (**h**) SGM, Mean/AD. (**i**) SGM, LoG/AD. (**j**) SGM, BilSub/AD. (**k**) SGM, Rank/AD. (**l**) SGM, Census.

**Figure 16 sensors-22-03332-f016:**
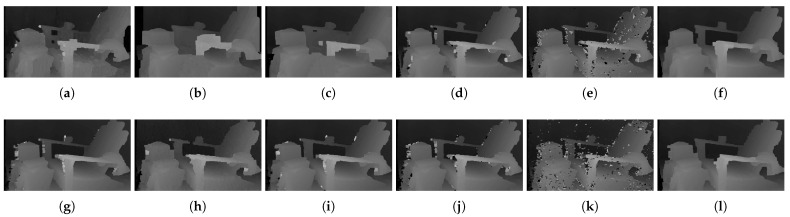
Estimated disparity images of the combinations of cost and image filter using the GC stereo method, according to the low and high *pattern contrast* parameters. The images in the top row show disparities at low contrast, and the images in the bottom row show those at high contrast, respectively. (**a**) GC, AD. (**b**) GC, Mean/AD. (**c**) GC, LoG/AD. (**d**) GC, BilSub/AD. (**e**) GC, Rank/AD. (**f**) GC, Census. (**g**) GC, AD. (**h**) GC, Mean/AD. (**i**) GC, LoG/AD. (**j**) GC, BilSub/AD. (**k**) GC, Rank/AD. (**l**) GC, Census.

**Figure 17 sensors-22-03332-f017:**
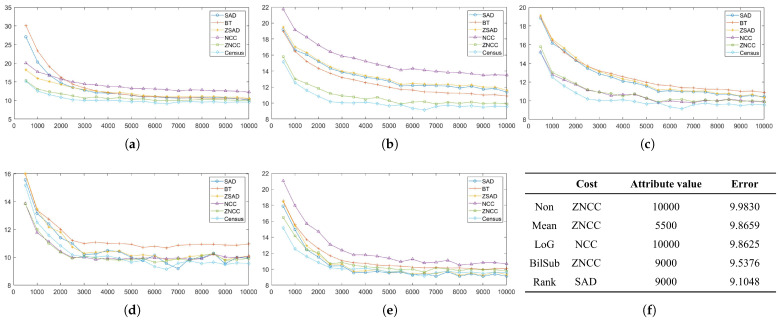
Errors of combinations of cost and image filter using the window-based stereo method, according to variations of the *number of pattern dots* parameter. (**a**) Non-filter. (**b**) Mean. (**c**) LoG. (**d**) BilSub. (**e**) Rank. (**f**) Best of Window.

**Figure 18 sensors-22-03332-f018:**
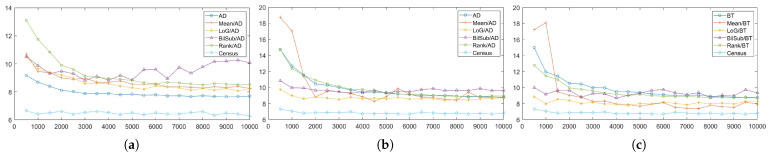
Errors of combinations of cost and image filter using the SGM and GC stereo methods, according to variations of the *number of pattern dots* parameter. (**a**) SGM/AD. (**b**) GC/AD. (**c**) GC/BT.

**Figure 19 sensors-22-03332-f019:**
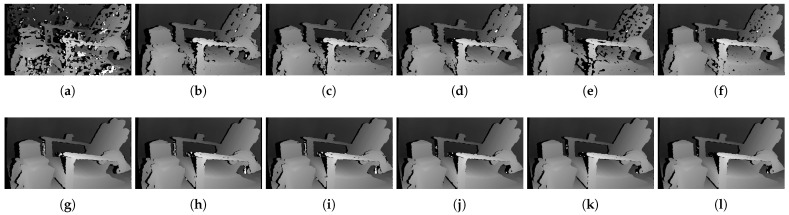
Estimated disparity images of the combinations of cost and image filter using the window-based stereo method, according to low and high parameters for the *number of pattern dots*. The images in the top row show disparities with the small *number of pattern dots*, and the images in the bottom row show those with the large *number of pattern dots*, respectively. (**a**) Window, SAD. (**b**) Window, Mean/SAD. (**c**) Window, LoG/SAD. (**d**) Window, BilSub/SAD. (**e**) Window, Rank/SAD. (**f**) Window, Census. (**g**) Window, SAD. (**h**) Window, Mean/SAD. (**i**) Window, LoG/SAD. (**j**) Window, BilSub/SAD. (**k**) Window, Rank/SAD. (**l**) Window, Census.

**Figure 20 sensors-22-03332-f020:**
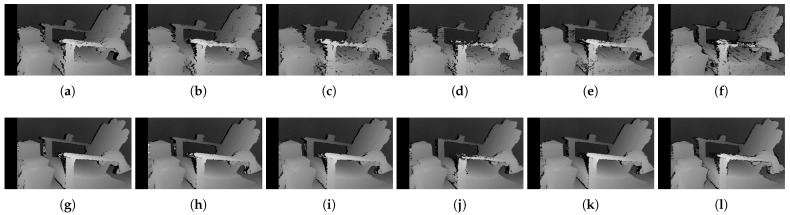
Estimated disparity images of the combinations of cost and image filter using the SGM stereo method, according to low and high parameters for the *number of pattern dots*. The images in the top row show disparities with the small *number of pattern dots*, and the images in the bottom row show those with the large *number of pattern dots*, respectively. (**a**) SGM, AD. (**b**) SGM, Mean/AD. (**c**) SGM, LoG/AD. (**d**) SGM, BilSub/AD. (**e**) SGM, Rank/AD. (**f**) SGM, Census. (**g**) SGM, AD. (**h**) SGM, Mean/AD. (**i**) SGM, LoG/AD. (**j**) SGM, BilSub/AD. (**k**) SGM, Rank/AD. (**l**) SGM, Census.

**Figure 21 sensors-22-03332-f021:**
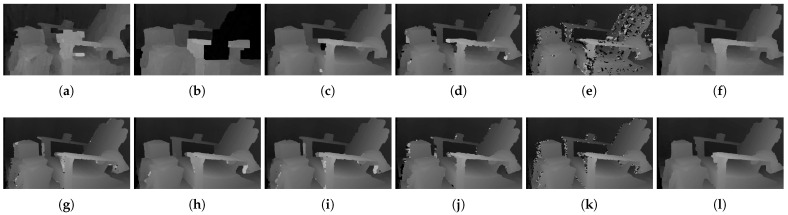
Estimated disparity images of the combinations of cost and image filter using GC stereo method according to low and high parameter of the *number of pattern dots*. The images in the top row show disparities with the small *number of pattern dots*, and the images in the bottom row show those with the large *number of pattern dots*, respectively. (**a**) GC, AD. (**b**) GC, Mean/AD. (**c**) GC, LoG/AD. (**d**) GC, BilSub/AD. (**e**) GC, Rank/AD. (**e**) GC, Census. (**g**) GC, AD. (**h**) GC, Mean/AD. (**i**) GC, LoG/AD. (**j**) GC, BilSub/AD. (**k**) GC, Rank/AD. (**l**) GC, Census.

**Figure 22 sensors-22-03332-f022:**
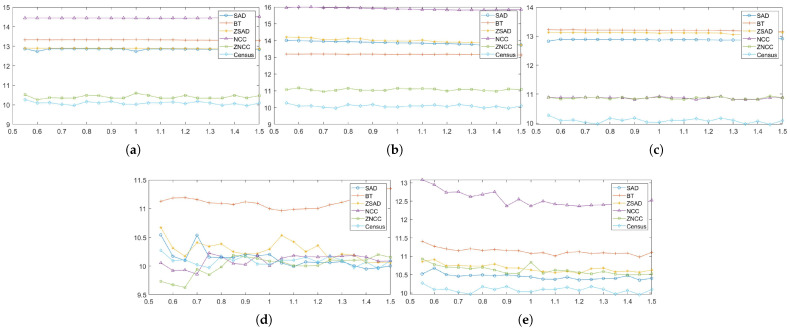
Errors of combinations of cost and image filter using the window-based stereo method, according to the variation of the image’s gain parameter. (**a**) Non-filter. (**b**) Mean. (**c**) LoG. (**d**) BilSub. (**e**) Rank.

**Figure 23 sensors-22-03332-f023:**
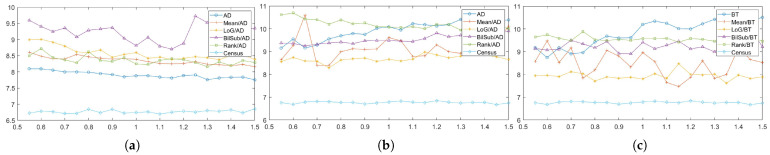
Errors of combinations of cost and image filter using the SGM and GC stereo methods, according to the variation of the image’s gain parameter. (**a**) SGM/AD. (**b**) GC/AD. (**c**) GC/BT.

**Figure 24 sensors-22-03332-f024:**
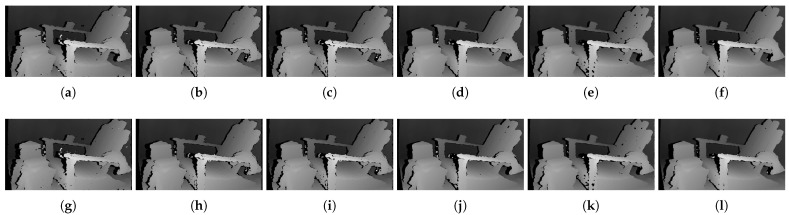
Estimated disparity images of the combinations of cost and image filter using the window-based stereo method, according to low and high gain parameters. The images in the top row show disparities at the low gain parameter, and the images in the bottom row show those at the high gain parameter, respectively. (**a**) Window, SAD. (**b**) Window, Mean/SAD. (**c**) Window, LoG/SAD. (**d**) Window, BilSub/SAD. (**e**) Window, Rank/SAD. (**f**) Window, Census. (**g**) Window, SAD. (**h**) Window, Mean/SAD. (**i**) Window, LoG/SAD. (**j**) Window, BilSub/SAD. (**k**) Window, Rank/SAD. (**l**) Window, Census.

**Figure 25 sensors-22-03332-f025:**
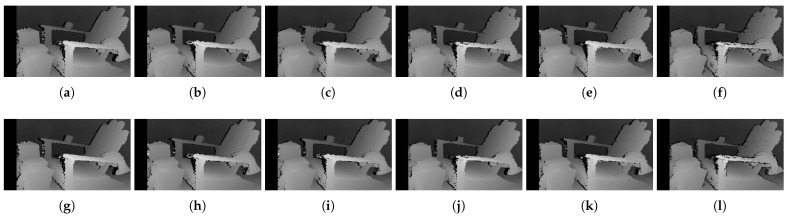
Estimated disparity images of the combinations of cost and image filter using the SGM stereo method, according to low and high gain parameter. The images in the top row show disparities at the low gain parameter, and the images in the bottom row show those at the high gain parameter, respectively. (**a**) SGM, AD. (**b**) SGM, Mean/AD. (**c**) SGM, LoG/AD. (**d**) SGM, BilSub/AD. (**e**) SGM, Rank/AD. (**f**) SGM, Census. (**g**) SGM, AD. (**h**) SGM, Mean/AD. (**i**) SGM, LoG/AD. (**j**) SGM, BilSub/AD. (**k**) SGM, Rank/AD. (**l**) SGM, Census.

**Figure 26 sensors-22-03332-f026:**
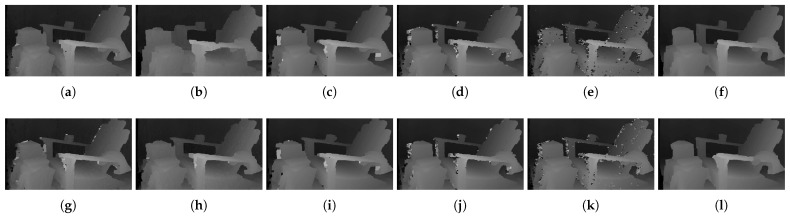
Estimated disparity images of the combinations of cost and image filter using the GC stereo method, according to low and high gain parameters. The images in the top row show disparities at the low gain parameter, and the images in the bottom row show those at the high gain parameter, respectively. (**a**) GC, AD. (**b**) GC, Mean/AD. (**c**) GC, LoG/AD. (**d**) GC, BilSub/AD. (**e**) GC, Rank/AD. (**e**) GC, Census. (**g**) GC, AD. (**h**) GC, Mean/AD. (**i**) GC, LoG/AD. (**j**) GC, BilSub/AD. (**k**) GC, Rank/AD. (**l**) GC, Census.

**Figure 27 sensors-22-03332-f027:**
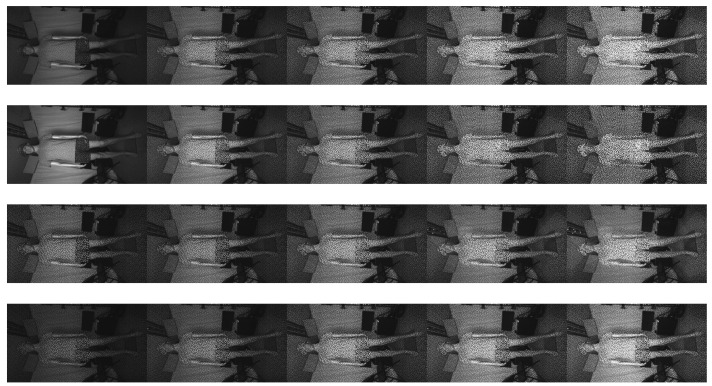
Left real active images, according to environments that match four attributes of active IR images. The figures show changes in the *pattern intensity, pattern contrast, number of pattern dots*, and *global gain* of IR images from top to bottom.

**Figure 28 sensors-22-03332-f028:**
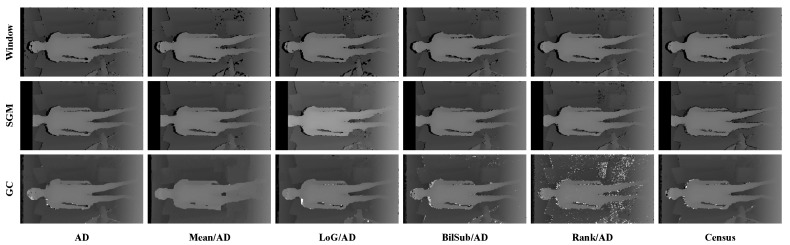
Estimated real disparity images of the combinations of cost and image filter using each stereo matching method. All attributes of active IR images were set to default values.

**Figure 29 sensors-22-03332-f029:**
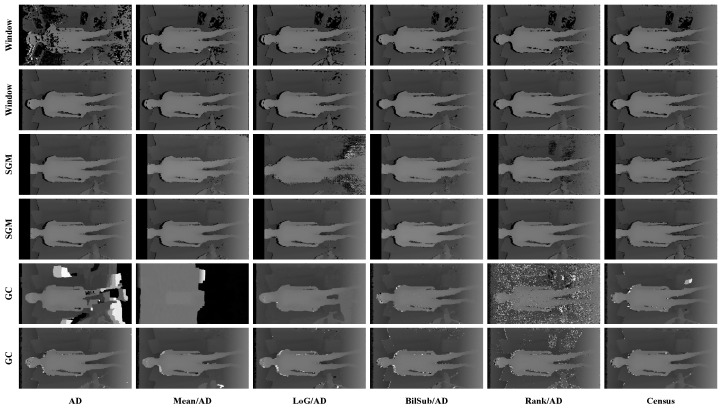
Estimated real disparity images of the combinations of cost and image filter using each stereo matching method, according to the *pattern intensity* parameter. The upper line indicates the low parameter, and the lower line indicates the high parameter for each stereo matching method, respectively.

**Figure 30 sensors-22-03332-f030:**
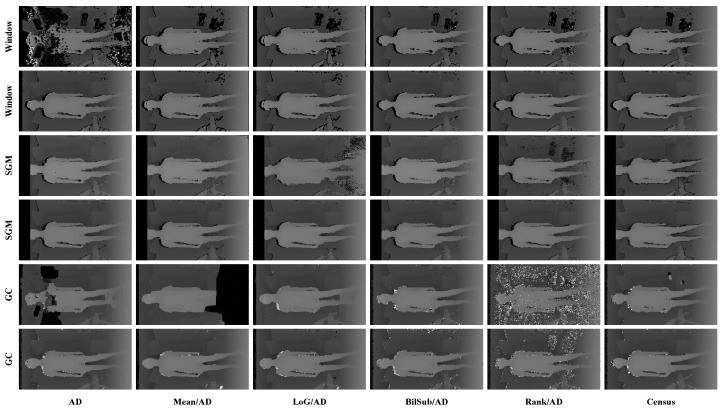
Estimated real disparity images of the combinations of cost and image filter using each stereo matching method, according to the *pattern contrast* parameter. The upper line indicates the low parameter, and the lower line indicates the high parameter for each stereo matching method, respectively.

**Figure 31 sensors-22-03332-f031:**
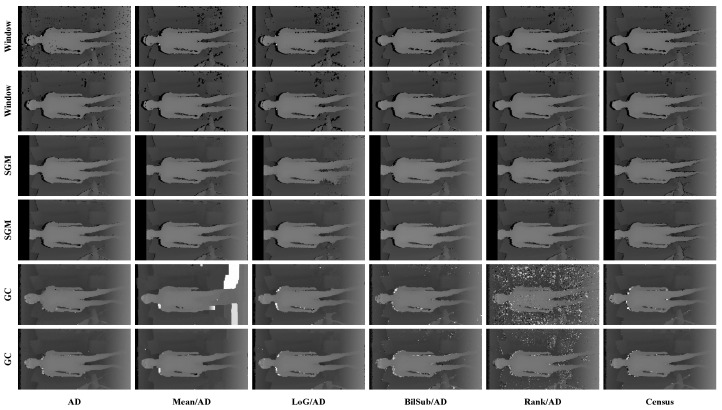
Estimated real disparity images of the combinations of cost and image filter using each stereo matching method, according to the parameter of the *number of pattern dots*. The upper line indicates the low parameter, and the lower line indicates the high parameter for each stereo matching method, respectively.

**Figure 32 sensors-22-03332-f032:**
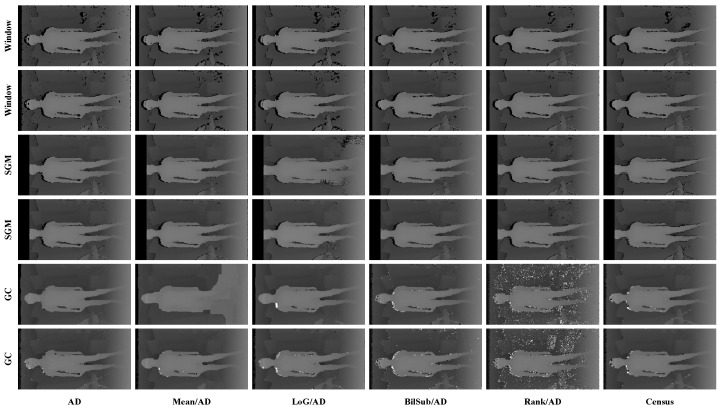
Estimated real disparity images of the combinations of cost and image filter using each stereo matching method, according to the gain parameter. The upper line indicates the low parameter, and the lower line indicates the high parameter for each stereo matching method, respectively.

**Table 1 sensors-22-03332-t001:** Specifications of commercial RGB-D cameras.

D400 Series Depth Cameras	D415	D435	D455
Depth module	D415	D430	D450
Baseline	55 mm	50 mm	95 mm
Left/Right Imagers Type	Standard	Wide	Wide
IR Projector	Standard	Wide	Wide
IR Projector FOV	H:67/V:41/D:75	H:90/V:63/D:99	H:90/V:63/D:99

**Table 2 sensors-22-03332-t002:** Runtime of stereo matching techniques using GPU for synthetic *Adirondack*(*s*).

	Window-Based						SGM			GC		
	SAD	BT	ZSAD	NCC	ZNCC	Census	AD	BT	Census	AD	BT	Census
Time (s)	0.0214	0.0237	0.0317	0.0261	0.0329	0.0316	0.453	0.465	0.505	10.321	10.326	10.438

**Table 3 sensors-22-03332-t003:** RMS errors of real active images (%).

Type	Laser Power					Illumination Ratio					The Number of Projectors				
Attribute Value	30	90	150	210	270	0.1	0.3	0.5	0.7	0.9	1	3	5	7	9
(W) Non/ZNCC	2.81	2.23	1.96	1.61	1.32	1.92	1.67	1.32	1.25	1.49	3.21	2.84	2.31	1.83	1.56
(W) Mean/ZNCC	3.27	2.98	2.42	1.96	1.41	2.14	1.63	1.58	1.71	1.83	2.98	2.66	2.20	1.84	1.45
(W) LoG/ZNCC	3.11	2.84	2.34	1.75	1.29	2.88	2.50	1.83	1.94	2.01	3.29	2.91	2.44	1.98	1.62
(W) BilSub/ZNCC	2.74	2.42	1.98	1.68	1.35	1.61	1.47	1.53	1.57	1.49	2.59	2.04	1.79	1.55	1.44
(W) Rank/SAD	2.94	2.71	2.12	1.56	1.33	2.09	1.84	1.56	1.61	1.66	3.75	2.91	2.28	1.87	1.67
(W) Census	1.71	1.58	1.41	1.29	1.31	1.99	1.71	1.57	1.45	1.61	2.71	2.47	1.89	1.58	1.41
(SGM) Census	0.96	0.94	1.08	1.03	0.99	1.06	0.95	1.10	0.99	1.11	1.07	1.03	0.98	0.95	1.01
(GC) Census	1.08	1.02	0.98	0.92	0.94	1.10	1.05	0.97	1.01	0.99	1.13	1.07	1.02	0.96	0.94

## Data Availability

Not applicable.

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
