# Peer review of "A Comparison and Evaluation of Stereo Matching on Active Stereo Images"

_sensors, 2022, doi:10.3390/s22093332_

Round 1

Reviewer 1 Report

In this paper, the authors analyzed and evaluated the relationship between the performance of active stereo technique and attributes of pattern texture using some synthetic and real active images. This topic is meaningful.  Extensive experiments are conducted in this work to evaluate the performance of different matching costs and optimization algorithms under the variations of pattern intensity, pattern contrast, the number of pattern dots, and global gain. 

However, in my opinion, the following aspects must be improved:

  1. The evaluation on the real active images in section 5 does not present quantitative results.
  2. The mean filter and LoG filter are used for preprocessing? Why put them under the title of "Matching Cost"?
  3. References: for example, the graph cut is not explicitly cited in section 3.2.2.
  4. Not much insights are provided. The authors give the experimental results without much analysis. For example, what components in an algorithm make it better than another one when facing the variation of a specific factor of pattern texture? I can only see a few related sentences in the conclusion: "Because SGM and GC use smoothness terms, the effect of texture-less region on images is not as significant as the window-based does. Consequentially, the purpose of pattern texture is lost. Thus, the performance of matching cost on SGM and GC is not affected by the attributes affecting the pattern texture.", which is not enough. 
  5. The English writhing needs to be polished. For example, "is influence by not only by" in line 9 should be "is influenced not only by", "various setting" in line 12 should be "various settings", "We generated synthetically active stereo images" in line 229 should be "We generated synthetic active stereo images", "Consequentially" in conclusion should be "Consequently".

Author Response

Response to Reviewer 1 Comments

In this paper, the authors analyzed and evaluated the relationship between the performance of active stereo technique and attributes of pattern texture using some synthetic and real active images. This topic is meaningful.  Extensive experiments are conducted in this work to evaluate the performance of different matching costs and optimization algorithms under the variations of pattern intensity, pattern contrast, the number of pattern dots, and global gain. 

However, in my opinion, the following aspects must be improved:

Thank you for reviewing our manuscript. We have carefully read your comments and revised our manuscript accordingly. The accordingly updated segments are written in blue bold fonts.

Point 1: The evaluation on the real active images in section 5 does not present quantitative results.

Response 1: Thank you for your keen comment. We used the Root Mean Square (RMS) error of Keselman, Leonid, et al. [1] to update quantitative experiments on real active images : (equation in the attached PDF file)

where z is best-fit plane value that is provided RealSense SDK [2],  z_i is estimated depth value of i-th pixel, and  is the number of pixels of plane position in images. We aligned a flat plane perpendicular to the off-the-shelf RGBD camera (i.e., RealSense D455 [3]). To express the four attributes that affect matching cost performance, we used factors of RealSense SDK and external environments such as the laser power, the illumination ratio of the laser power and the external light source, the number of IR projectors, and the illumination of the external light source. Overall quantitative results for real active images were not significantly different from those for synthetic active images.

We added tables, which lists the RMS error of combinations on the real data, and its description in section “5.1 Quantitative Results”.

[1] Keselman, L.; Iselin Woodfill, J.; Grunnet-Jepsen, A.; Bhowmik, A. Intel realsense stereoscopic depth cameras. Proceedings of the IEEE Conference on Computer Vision and Pattern Recognition Workshops, 2017, pp. 1–10.

[2] Corporation, I. Cross-platform library for Intel® RealSense™ depth cameras. https://github.com/IntelRealSense/librealsense.

[3] Grunnet-Jepsen, A.; Sweetser, J.N.; Woodfill, J. Best-known-methods for tuning Intel® RealSense™ D400 depth cameras for best performance. Intel Corporation: Satan Clara, CA, USA 2018, 1

Point 2: The mean filter and LoG filter are used for preprocessing? Why put them under the title of "Matching Cost"?

Response 2: Thank you for thoroughly reviewing our manuscript. We initially divided the sections based on based matching costs that compute difference either in raw intensity or in ordering of intensity. Since the filters also can be classified into these two categories, we described the filters with the matching costs because they belong in the same category. In addition, we structured our manuscript to consider two big components, which are matching costs and stereo algorithm, of active stereo matching affected by the four attributes we analyzed. We did not consider the filters as part of the combination to be tested but as an attribute affecting the result. However, after receiving comment and revising our manuscript once more, we found that the filters should be considered as part of the combination, and, thus, created a separate section for explaining filters. The creation of a new section allows us to solely analyze the effect of the four attributes.

Point 3: References: for example, the graph cut is not explicitly cited in section 3.2.2

Response 3: Thank you for your insightful feedback. We revised our manuscript to explicitly cite the source of any information that we missed.

Point 4: Not much insights are provided. The authors give the experimental results without much analysis. For example, what components in an algorithm make it better than another one when facing the variation of a specific factor of pattern texture? I can only see a few related sentences in the conclusion: "Because SGM and GC use smoothness terms, the effect of texture-less region on images is not as significant as the window-based does. Consequentially, the purpose of pattern texture is lost. Thus, the performance of matching cost on SGM and GC is not affected by the attributes affecting the pattern texture.", which is not enough. 

Response 4: Thank you for your helpful feedback. In this paper, we have evaluated how pattern intensity, pattern contrast, the number of pattern dots, and global gain affect matching cost performance. Through thorough experiments on four attributes affecting the performance of the active stereo technique, we separately analyzed and discussed the results produced by window and global method.

We revised our discussion and conclusion to include additional analysis and evaluation made. The runtime are measured and described in section “4.7 Comparison of Runtime”. The additional analysis can be found in section “6. Discussion and Conclusion”.

Point 5: The English writhing needs to be polished. For example, "is influence by not only by" in line 9 should be "is influenced not only by", "various setting" in line 12 should be "various settings", "We generated synthetically active stereo images" in line 229 should be "We generated synthetic active stereo images", "Consequentially" in conclusion should be "Consequently".

Response 5: We appreciate your careful review and valuable comments on our manuscript. After thorough revision, we have corrected grammatical errors in the manuscript.

Reviewer 2 Report

The authors investigated the influence of the attributes of pattern texture on active stereo matching performance. The paper was well-written. There are several comments from reviewer as follows: 

  1. The authors provided a wide range of performance evaluation on both synthetic and real dataset. What's about running time of different stereo methods? Is there any correlation between the attributes, e.g., pattern intensity, the number of pattern dots... on the running time? Reviewer encourage the authors to provide such the information in addition to quantitative and qualitative performance assessment. 
  2. in page 11, line 342-343, the authors stated that: "however, the errors....quickly saturates when applied with filters". However, it seems that the qualitative results in Fig. 7 does not really support this statement. Please elaborate this. 
  3. Reviewer do care about the saturation of error in this study. Any conclusion, for example, in which general value of pattern intensity, the error of most of matching costs will saturate? 
  4. This study investigated traditional stereo matching methods. However, there is a fact that most of the traditional methods are very low accuracy compared to the deep learning-based approach. To convince the audience,  Reviewer suggest that the authors should emphasize the importance of this study more in either the abstract or introduction section, even though the authors already stated this point at the end of conclusion section. 

Author Response

Response to Reviewer 2 Comments

The authors investigated the influence of the attributes of pattern texture on active stereo matching performance. The paper was well-written. There are several comments from reviewer as follows:

Thank you for reviewing our manuscript. We have carefully read your comments and revised our manuscript accordingly. The accordingly updated segments are written in blue bold fonts.

Point 1: The authors provided a wide range of performance evaluation on both synthetic and real dataset. What's about running time of different stereo methods? Is there any correlation between the attributes, e.g., pattern intensity, the number of pattern dots... on the running time? Reviewer encourage the authors to provide such the information in addition to quantitative and qualitative performance assessment.

Response 1: Thank you for your comment. We measured the running time of all combinations we tested in our experiment. The runtime is measured under default setting of pattern attributes in synthetic dataset. When comparing the runtime in terms of the matching algorithm, the window-based algorithm is faster than SGM and GC. The high runtime of the global method is caused by the optimization process of iteratively reducing the error. Due to high runtime, global method is not applicable to dynamic object reconstruction but to high quality static scene reconstruction. On the other hand, window method is more suitable to be applied to real-time applications, such as dynamic object reconstruction, even though it produces less accurate result than the global methods.

We have updated our manuscript to analysis of runtime in section ”4.7 Comparison of Runtime”.

Point 2: in page 11, line 342-343, the authors stated that: "however, the errors...quickly saturates when applied with filters". However, it seems that the qualitative results in Fig. 7 does not really support this statement. Please elaborate this.

Response 2: Thank you for thoroughly reviewing our manuscript. As you mentioned, we misinterpreted the results shown in Fig. 7. The convergence rate is slower when the filter is applied as shown in shown in Fig. 7.

We have revised our manuscript to correct the statement reviewer pointed out.

Point 3: Reviewer do care about the saturation of error in this study. Any conclusion, for example, in which general value of pattern intensity, the error of most of matching costs will saturate?

Response 3: Thank you for your insightful comment. According to our experimental results, the attributes that converge the error are the pattern intensity and the number of pattern dots. The laser power on IR projector of widely used off-the-shelf RGBD camera (i.e., RealSense D455 [1]) can be tuned from 0 to 360 in 30 steps. The default setting value of power provided by RealSense SDK is 150. We generated a synthetic active stereo dataset to replicate this value. Therefore, it can be said that the error at the default setting value of 150 in our experiment is sufficiently converged (Figs. 7). In addition, the experimental environment for the number of pattern dots is consistent with the environment for projecting patterns from multiple IR projectors. Multiple IR projectors project more pattern dots into the image than a single IR projector. Because of these denser dots, the error converges as the number of IR projectors emitting pattern dots increases. This trend can be confirmed in the quantitative section on additional experimental real data.

We have updated our manuscript to explain a synthetic active stereo dataset generation process in section “4.1 Dataset and Implementation” and quantitative experiments on real data in section “5.1 Quantitative Results”.

[1] Grunnet-Jepsen, A.; Sweetser, J.N.; Woodfill, J. Best-known-methods for tuning Intel® RealSense™ D400 depth cameras for best performance. Intel Corporation: Satan Clara, CA, USA 2018, 1.

Point 4: This study investigated traditional stereo matching methods. However, there is a fact that most of the traditional methods are very low accuracy compared to the deep learning-based approach. To convince the audience, Reviewer suggest that the authors should emphasize the importance of this study more in either the abstract or introduction section, even though the authors already stated this point at the end of conclusion section. 

Response 4: Thank you for your careful review. As the reviewer mentioned deep learning-based approaches may outperform traditional methods in many cases. However, the performance of deep learning-based methods is highly dependent on the dataset used to train the model. As a result, if the data given does not resemble any data in the training dataset, the performance of the deep learning-based method would most likely be low. Thus, the performance of the deep learning-based method is heavily dependent on the environment. In addition, since the deep learning-based method is dependent on the training data, it is difficult to analyze whether the performance evaluation result according to the pattern attributes is over-fitting the learning data or depends on the pattern attributes. In contrast, the traditional methods can be applied and analyzed generally.

Thus, regardless of the dataset, our experimental results on the attributes of pattern texture that affect the IR environment can provide guidance for constructing an active stereo sensor system for many applications of active stereo matching techniques.

The additional explanation with regards to our contribution is added to section “1. Introduction”.

Reviewer 3 Report

This paper provides a comprehensive survey of matching costs in active stereo with interesting results.

This paper is interesting, but there are a few unclear points that need to be corrected.

  1. Stereo-matching results can vary very sensitively depending on the parameters. For reproducibility, the paper should list all the stereo matching parameters or, if it would be very verbose, publish the entire source code.
  2. Matching is also highly dependent on how the random pattern is generated. In addition, the size of the pattern dots and their PSF are also important. Describe in detail how the projected pattern was generated in the simulation experiment at a reproducible level. Currently, it is not reproducible.
  3. For Figure 2, please use a high-resolution image without JPEG compression or include a Zoomed image to show the details of the projection. Currently, the projection pattern is broken by the JPEG encoder.
  4. Please list the CENSUS and RANK parameters; without the Windows size, we cannot identify the conversion.
  5. For the experiments in Figs. 5, 7, and 8, it is better to explain how to derive the various cost functions by filtering the input images in advance. It would be better to explain separately the method of zero averaging by filtering.
  6. In Figures 7 and 8, is there any significance in the intensity above 255, which exceeds the range of unsigned chars? Check once if the input data accepts more than 1 byte of data.
  7. For the results in Figures 7, 12, 17, and 22, we want the number of the best score to compare among the prefilters. Please write directly on the graph or make it as a table.
  8. In the experiments in Figures 12 and 13, the meaning of the pattern contrast parameter is unclear. Please define in detail. (Figure 12, you forgot the period.)

Author Response

Response to Reviewer 3 Comments

This paper provides a comprehensive survey of matching costs in active stereo with interesting results.

This paper is interesting, but there are a few unclear points that need to be corrected.

Thank you for reviewing our manuscript. We have carefully read your comments and revised our manuscript accordingly. The accordingly updated segments are written in blue bold fonts.

Point 1: Stereo-matching results can vary very sensitively depending on the parameters. For reproducibility, the paper should list all the stereo matching parameters or, if it would be very verbose, publish the entire source code.

Response 1: Thank you for your critical feedback. After thorough revision, we added the missing details of parameters for stereo matching algorithms in “4.1. Dataset and Implementation Details” section as follows:

The smoothness parameter of SGM is set as value × number of image channel × matched block size × matched block size, in StereoSGBM library of OpenCV (P_1=8x1x3x3, P_2=32x1x3x3 ) [1, 2]. For the smoothness parameter of GC, the max smooth value is 1 and the weight of smoothness term is 8000 in MRF library [3], respectively.

[1] Hirschmuller, H.; Scharstein, D. Evaluation of stereo matching costs on images with radiometric differences. IEEE Transactions on Pattern Analysis and Machine Intelligence 2008, 31, 1582–1599.

[2] Bradski, G.; Kaehler, A. Learning OpenCV: Computer vision with the OpenCV library; " O’Reilly Media, Inc.", 2008.

[3] Szeliski, R.; Zabih, R.; Scharstein, D.; Veksler, O.; Kolmogorov, V.; Agarwala, A.; Tappen, M.; Rother, C. A comparative study of energy minimization methods for markov random fields with smoothness-based priors. IEEE Transactions on Pattern Analysis and Machine Intelligence 2008, 30, 1068–1080.

Point 2: Matching is also highly dependent on how the random pattern is generated. In addition, the size of the pattern dots and their PSF are also important. Describe in detail how the projected pattern was generated in the simulation experiment at a reproducible level. Currently, it is not reproducible.

Response 2: Thank you for pointing out the missing details in our manuscript.

We generate a synthetic active stereo image from the input RGB image and depth image. Generally, the active pattern projected from the laser-based IR projector that follows the inverse square law is designed to emit random dots. Also, the laser speckle must appear on the object's surface.

To fully replicate the active pattern produced by the widely used off-the-shelf RGBD camera (i.e., RealSense D455 [4]) to our synthetic dataset, we empirically measure the size of an IR dot, which is a composite of the active pattern. We measured the size of the dot by projecting the pattern onto the fit plane from one meter distance away. From this environment, we discovered that a radius of each dot is 5mm. From the pair of RGB image and depth map, we first randomly sampled the location of each pattern dot using Kocis et. al.’s method [5]. After randomly sampling the location of each dot, we applied a 2D Gaussian kernel to each dot. The 2D Gaussian kernel is used a point spread function (PSF) to generate laser speckle for each dot. Then, we applied inverse square law to adjust the size and intensity of each dot based on depth. We finetuned the parameter for the 2D Gaussian kernel and inverse square law so that the size of a dot is 5mm when it is 1 meter away.

Then, the RGB image is converted to a gray-scale image, assuming that the receiving wavelength of the IR sensor is encoded with the intensity of the monochrome imaging sensor converted in the RGB camera. Finally, the gray-scale image and the random patterns are integrated to generate a synthetic active stereo image.

We updated the manuscript to include the details of generating synthetic data can be found in “4.1. Dataset and Implementation Details” section.

[4] Grunnet-Jepsen, A.; Sweetser, J.N.; Woodfill, J. Best-known-methods for tuning Intel® RealSense™ D400 depth cameras for best performance. Intel Corporation: Satan Clara, CA, USA 2018, 1

[5] Kocis, L., and W. J. Whiten. “Computational Investigations of Low-Discrepancy Sequences.” ACM Transactions on Mathematical Software. Vol. 23, No. 2, 1997, pp. 266–294.

Point 3: For Figure 2, please use a high-resolution image without JPEG compression or include a Zoomed image to show the details of the projection. Currently, the projection pattern is broken by the JPEG encoder.

Response 3: Thank you for noticing the mistakes we made. We have replaced our image in Figure 2 with high-resolution image to convey more details.

Point 4: Please list the CENSUS and RANK parameters; without the Windows size, we cannot identify the conversion

Response 4: Thank you for your careful review. The filter size of Census and Rank is 9 × 9. The filer size is tuned heuristically to achieve the best performance at the default values of four attributes of active IR images such as pattern intensity, pattern contrast, the number of pattern dots, and global gain. We have included the details of Census and Rank parameters in “4.1. Dataset and Implementation Details” section of our manuscript.

Point 5a: For the experiments in Figs. 5, 7, and 8, it is better to explain how to derive the various cost functions by filtering the input images in advance.

Response 5a: Thank you for your insightful comment. To thoroughly evaluate the matching performance, we have tested all possible combinations of filters, matching costs, and stereo matching algorithms. While all filters can be used regardless of the matching algorithm, window-based matching costs (SAD, ZSAD, NCC, and ZNCC) cannot be implemented with semi-global method (SGM) and graph-cut (GC). This is because SGM optimizes the disparity values along 1D path in eight directions, and GC optimizes the disparity map error globally. Because they are optimizing the disparity value globally or in 1D path, it is not suitable to use the window-based matching cost. Thus, due to implementation difference, window-based matching costs cannot be applied to SGM and GC methods. On the other hand, window-based algorithm can aggregate all possible combination of matching costs and filters. We revised the manuscript to include these descriptions of how to derive the cost functions in the experimental section (Sec. 4).

Point 5b: It would be better to explain separately the method of zero averaging by filtering.

Response 5b: Thank you for your comment. We reorganized our manuscript via creating new sections (3.2.a and 3.2.e) to explain the methods using zero averaging filters.

Point 6: In Figures 7 and 8, is there any significance in the intensity above 255, which exceeds the range of unsigned chars? Check once if the input data accepts more than 1 byte of data.

Response 6: Thank you for thoroughly reviewing our manuscript. We evaluated each combination at 60 levels (from 5 to 300) that specifies the magnitude of the 2D Gaussian kernel. The pattern texture of our synthetic active stereo image is generated using a 2D Gaussian kernel that mimics a laser speckle. Following the inverse square law, the intensity of each pattern dots depends on the distance between the surface and camera, which is depth. Further the surface, lower the intensity of pattern is set. Thus, the level of pattern intensity can go over 255 and is significant for our experiment. Even though pattern dots, which are close to camera, already reach the intensity of 255, pattern dots on far surface still have room for increase. Thus, we tried to increase the intensity of pattern dots far from the camera.

We updated our manuscript accordingly to include this explanation. The update section can be found in section “4.3 Evaluation on Pattern Intensity Changes”

Point 7: For the results in Figures 7, 12, 17, and 22, we want the number of the best score to compare among the prefilters. Please write directly on the graph or make it as a table.

Response 7: Thank you for your thoughtful feedback. We added the tables, which lists the best performing matching costs and their scores for Figures 7, 12, and 17. The table did not consider Census because we did not use Census on filtered images because Census applies its own filter on image when computing the matching cost. We did not add the table for Figure 22 because there is no significant result that the table can show separately.

Point 8: In the experiments in Figures 12 and 13, the meaning of the pattern contrast parameter is unclear. Please define in detail. (Figure 12, you forgot the period.)

Response 8: Thank you for your comment. The pattern contrast is the intensity ratio between the IR  image and the pattern image. The pattern contrast is set in the last stage of generating a synthetic active stereo image. The x-axis in Figs. 12 and 13 means 20 levels (from 0.05 to 0.1) of the intensity of pattern image relative to the intensity of IR image. For example, if the relative intensity of pattern image is set as 0.05, the relative intensity of the IR image is set as 0.95. We analyzed how the passive texture, which is the original IR image excluding pattern dots, and the pattern texture, which is pattern dots projected onto the image, affect each other on stereo matching. To analyze the disparity estimation accuracy for the passive texture and pattern texture intensity ratio, we measure the error by changing the intensity ratio of the pattern image and the original IR image excluding pattern dots. Figs. 12 and 13 show the quantitative results about pattern contrast change experiments. This shows that the disparity estimation is not accurate not only when the intensity of the pattern texture is low, but also when the passive texture is too dark. Therefore, it is necessary to find the optimal pattern contrast level for accurate disparity estimation.

We added the meaning of the pattern contrast and analysis of experiments about contrast changes in section “4.4 Evaluation on Contrast Changes”.

Round 2

Reviewer 1 Report

All my concerns have been carefully addressed by the authors, especially new results on real active images, more discussions, and the re-structuring of the manuscript. 

A minor point: line 643, fig. ??

Author Response

Response to Reviewer 1 Comments

All my concerns have been carefully addressed by the authors, especially new results on real active images, more discussions, and the re-structuring of the manuscript. 

A minor point: line 643, fig. ??

Response 1: Thank you for your careful review of our manuscript. We have updated our manuscript manuscript accordingly to correct the mistake you pointed out. The updated sentence is written in blue bold font.

Reviewer 2 Report

Thank you for revising the manuscript !!!

Author Response

Response to Reviewer 2 Comments

Thank you for revising the manuscript !!!

Response: Thank you so much for thoroughly reviewing our manuscript. With your help, we were able to significantly improve our manuscript.
